# AMOTL2 mono-ubiquitination by WWP1 promotes contact inhibition by facilitating LATS activation

Daehee Hwang[1] , Miju Kim[1], Soyeon Kim[2], Mi Ra Kwon[2], Ye-Seul Kang[1], Dahyun Kim[1], Ho-Chul Kang[2], Dae-Sik Lim[1]

Contact inhibition is a key cellular phenomenon that prevents cells from hyper-proliferating upon reaching confluence. Although not fully characterized, a critical driver of this process is the Hippo signaling pathway, whose downstream effector yes-associated protein plays pivotal roles in cell growth and differentiation. Here, we provide evidence that the E3 ligase WWP1 (WW-domain containing protein 1) mono-ubiquitinates AMOTL2 (angiomotin-like 2) at K347 and K408. Mono-ubiquitinated AMOTL2, in turn, interacts with the kinase LATS2, which facilitates recruitment of the upstream Hippo pathway component SAV1 and ultimately promotes yes-associated protein phosphorylation and subsequent cytoplasmic sequestration and/or degradation. Furthermore, contact inhibition induced by high cell density promoted the localization and stabilization of WWP1 at cell junctions, where it interacted with Crumbs polarity proteins. Notably, the Crumbs complex was functionally important for AMOTL2 mono-ubiquitination and LATS activation under high cell density conditions. These findings delineate a functionally important molecular mechanism in which AMOTL2 mono-ubiquitination by WWP1 at cell junctions and LATS activation are tightly coupled to upstream cell density cues.

## Introduction

Cells stop proliferating once they have reached an appropriate density, a phenomenon known as contact inhibition (Eagle et al, 1965). Although the loss of contact inhibition occurs in rapidly growing normal tissues during embryonic development and regeneration (McClatchey & Yap, 2012), it is nevertheless one of the hallmarks of many cancers and developmental anomalies (Hanahan & Weinberg, 2000). Therefore, understanding the molecular basis for regulation of contact inhibition is of paramount importance, not only for clinical purposes but also from an evolutionary perspective, as it distinguishes metazoans from unicellular organisms, whose growth potential is limited only by surrounding nutrients (Mayor & Carmona-Fontaine, 2010).

The Hippo signaling pathway has been recognized as a critical hub for orchestrating various developmental programs in response to diverse extracellular stimuli, including cell-cell contact (Zhao et al, 2007; Gumbiner & Kim, 2014), actomyosin tension (Zhao et al, 2012; Aragona et al, 2013), energy stress (Mo et al, 2015; Wang et al, 2015), and G-protein–coupled receptor ligands such as lysophosphatidic acid (LPA) (Yu et al, 2012). In fact, the field is becoming increasingly complex, reflecting a growing appreciation of crosstalk with other signaling pathways (Yu et al, 2015). Activation of the canonical Hippo pathway in mammals is mediated by a kinase cascade initiated by MST1/2 (mammalian sterile 20-like kinase-1 and -2) and its cofactor SAV1 (salvador family WW domain-containing protein 1), which phosphorylate LATS1 (large tumor suppressor kinase-1) on Thr-1079 (or Thr-1041 for LATS2) within the hydrophobic motif (Cordenonsi et al, 2011; Yin et al, 2013). Activated LATS1/2 then phosphorylate the transcriptional coactivator (YAP) yes-associated protein on numerous residues, with Ser-127 phosphorylation mediating YAP/TAZ interaction with 14-3-3 proteins and primarily governing YAP/TAZ cytoplasmic sequestration and Ser-381 phosphorylation promoting proteolytic degradation (Dong et al, 2007). In contrast, YAP de-phosphorylation promotes its translocation to the nucleus, where it binds to members of the TEAD family of transcription factors to up-regulate a number of target genes involved in stem/progenitor cell expansion and tissue homeostasis (Mo et al, 2014; Kim et al, 2015a, 2015b).

Multiple independent reports have shown that angiomotin family proteins are potent suppressors of YAP (Chan et al, 2011; Paramasivam et al, 2011; Wang et al, 2011; Zhao et al, 2011), although some groups claim these proteins have oncogenic capacities in certain contexts (Ranahan et al, 2011; Yi et al, 2013). Angiomotins, which include angiomotin (AMOT p80 and p130 isoforms), angiomotin-like 1 (AMOTL1), and angiomotin-like 2 (AMOTL2), inactivate YAP by sequestering it within the cytoplasm (Chan et al, 2011; Zhao et al, 2011) and/or activating LATS kinase by acting as a scaffold for other membrane-bound Hippo pathway components, including PALS1 (protein associated with Lin-7 1), PATJ (Pals1-associated tight junction), and NF2 (neurofibromatosis type 2) (Wells et al, 2006; Varelas et al, 2010). We have previously shown that the latter mechanism operates, at least in part, through regulation of AMOTL2 mono-ubiquitination at K347 and K408 by the de-ubiquitinase, USP9X (Kim et al, 2016). This provides a docking site for LATS binding, which is mediated by its ubiquitin

[1]National Creative Research Center for Cell Plasticity, Department of Biological Sciences, Korea Advanced Institute of Science and Technology, Daejeon, Korea   [2]Genomic Instability Research Center, Ajou University School of Medicine, Suwon, Korea

Correspondence: daesiklim@kaist.ac.kr
Miju Kim's present address is Department of Medical Oncology, Dana-Farber Cancer Institute, Harvard Medical School, Boston, MA, USA
Ye-Seul Kang's present address College of Medicine, Yonsei University, Seoul, Korea

(Ub)-associated domain (UBA). However, the specific contexts in which this ubiquitination event takes place and how LATS activity becomes strengthened by its interaction with AMOTL2 are not known.

To answer these remaining questions, we performed a small-scale manual in vivo ubiquitination screen using an ORF library comprising 160 E3 ligases. This screen identified the HECT-type E3 ligase WWP1 (WW-domain containing protein 1) as being responsible for AMOTL2 mono-ubiquitination, a modification that readily promoted LATS activation. WWP1-mediated mono-ubiquitination of AMOTL2 not only enabled AMOTL2-LATS2 interactions, it also facilitated recruitment of the up-stream Hippo pathway component SAV1. Mechanistically, contact inhibition induced by high cell density promoted the localization and stabilization of WWP1 at cell–cell junctions. WWP1 colocalized with the Crumbs complex, which is crucial in establishing apical-basal polarity in epithelial cells (Roh et al, 2003), and this targeting was achieved through direct interactions with AMOTL2. Collectively, our findings provide novel molecular insights into a potential mechanism by which cell–cell contact activates the Hippo pathway, a process that is in part orchestrated by the mono-ubiquitination of AMOTL2 by WWP1.

# Results

## WWP1 E3 ligase directly interacts with and mono-ubiquitinates AMOTL2

We have previously characterized the role of USP9X as a de-ubiquitinase of AMOTL2 (Kim et al, 2016)—a scaffolding protein critical in the mobilization of Hippo pathway components, especially at cell-cell junctions. However, the detailed molecular mechanisms underlying Ub-AMOTL2-mediated LATS activation and the specific upstream context in which this ubiquitination event is favored were not fully addressed. To answer these questions, we first sought to identify the E3 ligase that presumably functions in opposition to USP9X to mono-ubiquitinate AMOTL2 at K347/K408. To this end, we performed in vivo ubiquitination assays on AMOTL2 using an E3 ligase ORF library comprising ~160 E3 ligases (Table S1). From this initial round of screening, we identified 29 E3 ligases which appeared to display ubiquitination activity on AMOTL2 (Fig S1A and Table S2). Upon closer analysis of only the E3 ligases which exhibited the strongest ubiquitination effect (as well as ITCH, which has independently been reported to display ubiquitination activity on AMOT [Wang et al, 2012a]), the HECT-type E3 ligase WWP1 was the prime candidate which potently mono-ubiquitinated WT AMOTL2, but not the mono-ubiquitination-defective K347R/K408R (2KR) AMOTL2 mutant (Figs 1A and S1B). As anticipated, treatment with heclin, a HECT type–specific E3 ligase inhibitor (Mund et al, 2014), prevented AMOTL2 mono-ubiquitination (Fig S2A). Moreover, the catalytically inactive C890A WWP1 mutant failed to ubiquitinate AMOTL2, thus confirming that the ubiquitination event is direct, and not a non-specific byproduct (Fig 1B).

To directly assess this mono-ubiquitination event and to rule out the possibility that this phenomenon is a non-specific bystander effect, we purified full-length AMOTL2 and WWP1 proteins from baculovirus-infected Sf9 insect cells and performed in vitro ubiquitination. An identical, single ubiquitination band was observed using either WT Ub or a lysine-free, non-polymerizable K0 Ub mutant, whereas mono-ubiquitination of AMOTL2 2KR was dramatically reduced (Fig 1C).

We note that residual ubiquitination of AMOTL2 2KR by WWP1 exists presumably because of the presence of other, yet unidentified mono-ubiquitination sites that are independent from regulation by USP9X. These sites likely reside outside the coiled coil domain, because this domain was selected specifically for the identification of ubiquitination sites in our earlier study (Kim et al, 2016). Collectively, these results indicate that the major mode of AMOTL2 ubiquitination by WWP1 is mono-ubiquitination. Importantly, USP9X depletion increased AMOTL2 mono-ubiquitination, an effect that was abolished by co-depletion of WWP1 using two independent small hairpin RNAs (shRNAs), confirming that the two enzymes indeed exhibit contrasting functions on the same AMOTL2 lysine residues (Fig S2B).

Next, to assess whether direct physical binding is required for AMOTL2 ubiquitination by WWP1, we performed a series of co-immunoprecipitation assays. We found that purified GST-WWP1 and AMOTL2-V5 proteins interact in pull-down assays in vitro (Fig 2A) and that the native proteins also interact at endogenous levels in vivo (Fig 2B). In line with these findings, a recent screening study from an independent group found that AMOT and AMOTL1 proteins interact strongly with WWP1, as well as other HECT-type E3 ligases, such as ITCH and NEDD4L (Vargas et al, 2020).

To further characterize the direct interaction between WWP1 and AMOTL2, we performed a domain mapping analysis. WWP1 contains as many as four tandem WW domains, an N-terminal C2 domain critical for plasma membrane localization, and a C-terminal HECT domain required for its catalytic activity (Chen et al, 2017) (Fig 2C). For its part, AMOTL2 contains two L/PPxY-like motifs, 104-LPTY-107 and 210-PPQY-213 (Moleirinho et al, 2014). Accordingly, we reasoned that AMOTL2–WWP1 binding could be mediated by WW–L/PPxY interactions. As expected, deletion of the WW domains in WWP1 ablated interactions with AMOTL2 (Fig 2D). Interestingly, whereas singly mutating Tyr-107 (1m) or Tyr-213 (2m) in AMOTL2 had little or no noticeable effect on binding, mutation of both residues (1 + 2m) completely abrogated AMOTL2 interactions with WWP1 (Fig 2E), suggesting that the two L/PPxY motifs play redundant roles in mediating WWP1 binding. Furthermore, WWP1 efficiently mono-ubiquitinated other full-length angiomotin family members, including AMOT (p130) and AMOTL1 (Fig S3A and B), which also contain coiled-coil domains that harbor the conserved mono-ubiquitination sites. Functionally speaking, binding of WWP1 to AMOTL2 appeared to be important, as the 1 + 2m mutant failed to be mono-ubiquitinated upon the overexpression of WWP1 (Fig S4A). In addition, whereas wild-type WWP1 was efficiently pulled down by YAP, WW domain-deleted WWP1 mutant was not (Fig S4B), and conversely, WW domain-deleted YAP failed to pull down WWP1 (Fig S4C). These results imply that WWP1 constitutes a complex that includes even the most downstream Hippo signaling elements. Collectively, these findings indicate that WWP1 directly mono-ubiquitinates AMOTL2 at K347/K408 (and presumably AMOT and AMOTL1 at conserved corresponding lysine residues), a modification that is mediated by direct WW–L/PPxY interactions.

## WWP1 promotes LATS activation and thereby inhibits YAP nuclear localization and transcriptional activation

Based on the results from our previous study, we next tested whether increased mono-ubiquitination of AMOTL2 by WWP1 activation leads to an increase in the association of AMOTL2 with LATS2, and thereby promotes LATS activation and subsequent YAP inactivation. Indeed,

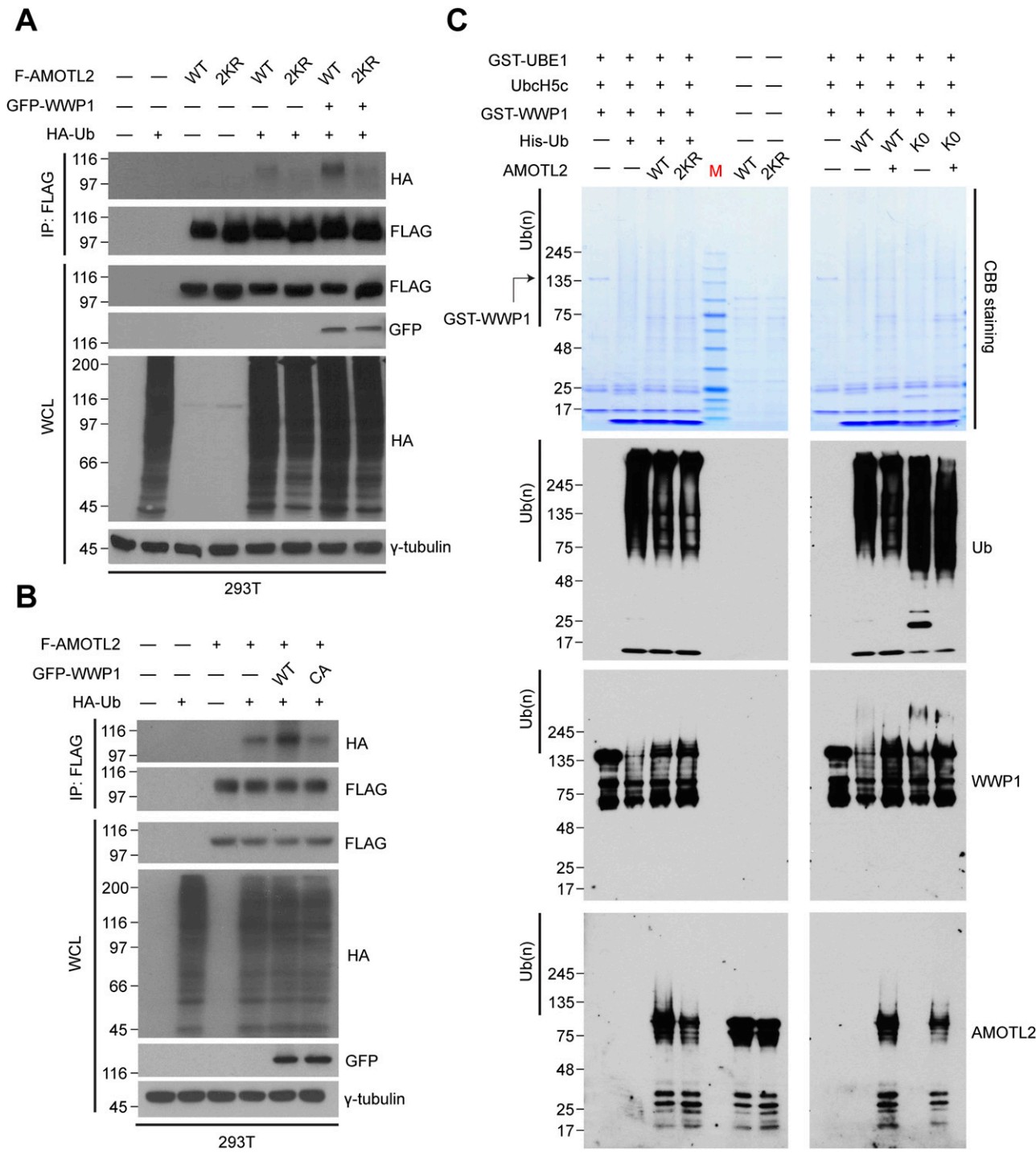

**Figure 1. AMOTL2 is a mono-ubiquitination substrate of the E3 ligase WWP1.**
**(A)** 293T cells were transfected with the indicated DNAs, then subjected to in vivo ubiquitination assays. **(B)** 293T cells were transfected with the indicated DNAs, then subjected to in vivo ubiquitination assays. **(C)** GST-WWP1 and AMOTL-V5 proteins purified from baculovirus were mixed with GST-UBE1, UbcH5c, and His-Ub as indicated and subjected to in vitro ubiquitination assay. M (molecular weight marker lane), CBB (Coomassie Brilliant Blue).

stable expression of FLAG-WWP1 in sparsely cultured MCF10A cells led to an increase in the levels of Ser-127–phosphorylated YAP (p-YAP) (Fig 3A). Importantly, this event was rescued in cells expressing the catalytically inactive WWP1 C890A mutant, demonstrating that the enzymatic activity of WWP1, rather than a scaffolding function, is

responsible for LATS activation. Consistent with this, notable YAP target genes, including *CTGF*, *CYR61*, *ANKRD1*, and *SKP2* (Jang et al, 2017), were markedly decreased in cells overexpressing WT WWP1, but not in those overexpressing C890A mutant WWP1 (Fig 3B). In line with these data, there was a dramatic reduction of YAP in the nuclear fraction

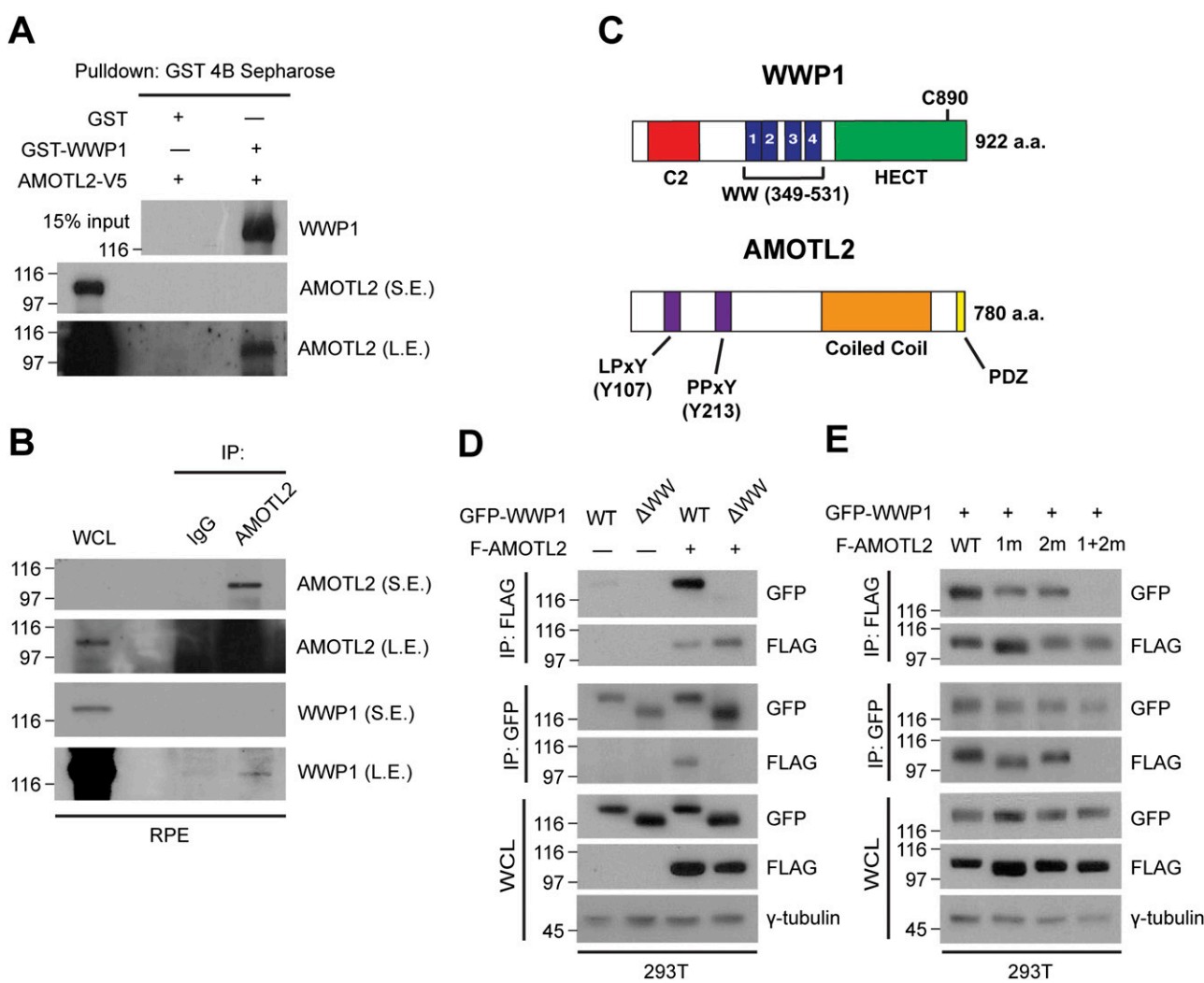

**Figure 2.  WWP1 physically interacts with AMOTL2.**
**(A)** GST-WWP1 and AMOTL-V5 proteins purified from baculovirus were subjected to in vitro binding assays using GST-4B Sepharose to pull down GST or GST-WWP1. S.E., short exposure; L.E., long exposure. **(B)** Confluent RPE-1 cells were lysed and endogenous proteins were co-immunoprecipitated with an anti-AMOTL2 antibody. **(C)** Schematic diagram illustrating the functional domains of WWP1 and AMOTL2. The amino acids shown in brackets represent sites either deleted or point-mutated in WWP1 and AMOTL2, respectively. **(D)** 293T cells were transfected with the indicated DNAs then subjected to reciprocal co-immunoprecipitation assays. **(E)** 293T cells were transfected with the indicated DNAs then subjected to reciprocal co-immunoprecipitation assays.

upon overexpression of WT WWP1 in sparsely cultured MCF10A cells (Fig 3C). Conversely, depletion of WWP1 in confluent MCF10A cells resulted in a decrease in p-YAP (Fig 3D), and a corresponding increase in YAP target gene expression (Fig 3E). In line with this, nuclear YAP was increased in WWP1-depleted cells cultured at confluence as assessed by nuclear-cytoplasmic fractionation (Fig 3F) and immunostaining for YAP (Fig 3G and H). Similar results were obtained in (RPE-1) retinal pigment epithelial-1 cells (Fig S5A–F). In agreement with these data, we noted an increase in cellular proliferation upon WWP1 depletion as assessed by BrdU incorporation, which was rescued by the re-introduction of shRNA-resistant wild-type WWP1, but not the C890A mutant (Fig S6A–C). Notably, the WW domain-deleted WWP1 mutant also failed to inactivate YAP (Fig S4D), suggesting that both the catalytic activity of WWP1 and direct physical binding between WWP1 and AMOTL2 are important for regulation of YAP activity. To further corroborate the above results, we performed a LATS2 in vitro kinase assay in the presence of ubiquitin using maltose-binding

protein (MBP) tagged YAP as a substrate. As expected, co-expression of AMOTL2 and LATS readily enhanced YAP phosphorylation. However, this spike in YAP phosphorylation was greatly attenuated by WWP1 depletion, an effect that was reversed by re-introduction of WT WWP1, but not C890A mutant WWP1 (Fig 3I). Importantly, the increase in YAP phosphorylation by WWP1 was indeed mediated by LATS, as assessed by the increase in LATS phosphorylation at the hydrophobic motif upon WWP1 overexpression (Fig S7A and B). Collectively, these data indicate that WWP1 overexpression facilitates LATS activation, leading to the nuclear exclusion of YAP and inhibition of its activity.

It is worth noting that a previous study has reported a role for WWP1 in poly-ubiquitinating LATS1, thereby causing its proteasomal degradation (Yeung et al, 2013). Thus, in this case WWP1 promoted YAP activity, resulting in an increase in cell proliferation, with the converse observed for cells depleted of WWP1. However, under our experimental conditions LATS1/2 protein levels remained unaltered in cells expressing WWP1,

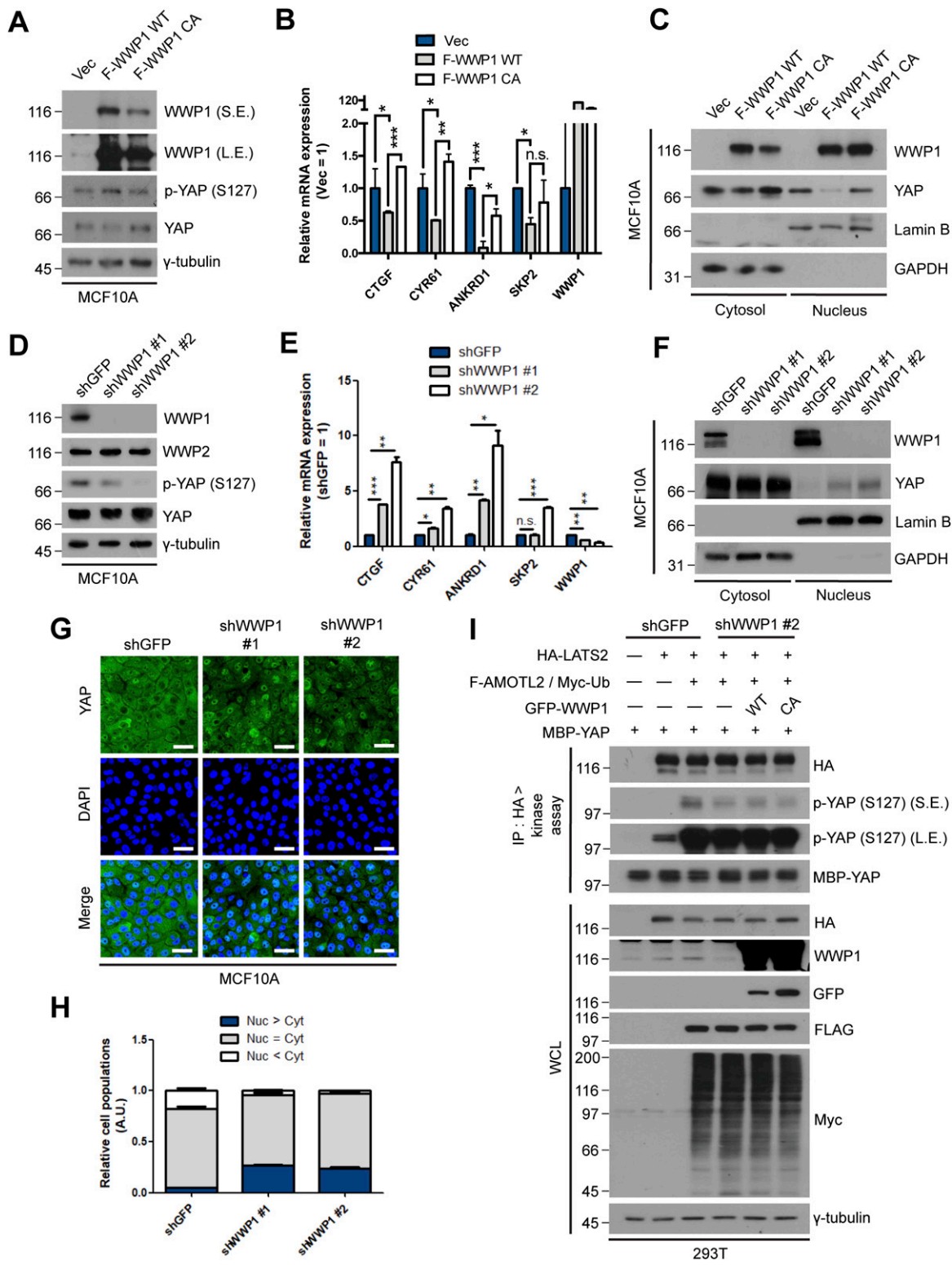

**Figure 3. WWP1 restrains YAP nuclear localization and suppresses its transcriptional activity.**
**(A)** MCF10A cells were stably transduced with control, FLAG-WWP1 WT, or C890A mutant retroviruses, re-seeded at sparse conditions, then analyzed for the indicated proteins by Western blotting. **(A, B)** mRNAs extracted from cells prepared as in (A) were analyzed for the indicated genes by qRT-PCR, and expression levels were normalized to those of GAPDH mRNA (n = 4). Data are expressed as means ± SEM (error bars; *P < 0.05, **P < 0.01, ***P < 0.001; unpaired t test). **(A, C)** Cells prepared as in (A) were fractionated into cytosolic and nuclear extracts, and analyzed by Western blotting for the indicated proteins. GAPDH and lamin B serve as controls for the cytosolic and nuclear fractions, respectively. **(D)** MCF10A cells stably transduced with shGFP (control) or shWWP1 were seeded under confluent conditions,, then lysed and analyzed

suggesting that the functional role WWP1 adopts may be context-dependent (Fig S7C and D). For instance, we were only able to consistently observe YAP inhibition upon WWP1 overexpression in sparsely cultured cells. In any case, the underlying point is that diverse, complex intracellular mechanisms are required to achieve the fine regulation of WWP1 activity necessary to ensure homeostasis.

## WWP1 promotes LATS-mediated AMOTL2 phosphorylation

In addition to YAP and TAZ, AMOTL2 has also been shown to be a target of LATS2, which phosphorylates AMOTL2 residue Ser-159 (Adler et al, 2013; Chan et al, 2013; Dai et al, 2013). This event physically occludes the binding interface between AMOTL2 and filamentous actin (F-actin), thereby further enabling the interaction of AMOTL2 with YAP. Consequently, AMOTL2 sequesters YAP within the cytosol and, by virtue of their physical association, inhibits YAP-TEAD transcriptional activity. On the basis of these observations, we investigated whether LATS2 activation induced by overexpression of WWP1 ultimately promotes the phosphorylation of AMOTL2. In line with our earlier findings, WWP1 depletion decreased AMOTL2 phosphorylation (Fig 4A); conversely, overexpression of WT WWP1, but not AMOTL2 binding–defective ΔWW mutant WWP1, increased AMOTL2 phosphorylation (Fig 4B). To our surprise, however, WWP1 also promoted the phosphorylation of AMOTL2 2KR mutant (Fig 4C), and the UBA domain-deleted LATS promoted AMOTL2 phosphorylation to a level comparable to wild-type LATS (Fig S8A). These results hint at the possibility that there may exist another molecular pathway in which WWP1 promotes LATS activation independent from its association with AMOTL2. In any case, WWP1 overexpression did not increase phosphorylation of the AMOTL2 S159A mutant (Fig S8B), and AMOTL2 phosphorylation was abolished in lysates from LATS1/2-deleted cells (Fig S8C), validating the specificity of the anti-phospho antibody used. Last, to establish a cause–effect relationship among these events, we used a genetic approach, testing whether LATS2 activation, and consequently AMOTL2 phosphorylation, occurs after AMOTL2 mono-ubiquitination by performing in vivo ubiquitination assays in LATS1/2-knockout cells. As can be seen, WWP1 overexpression-induced AMOTL2 mono-ubiquitination was not impaired in LATS-deleted cells (Fig 4D). In addition, mono-ubiquitination of the AMOTL2 S159A mutant was unhindered (Fig S8D), which establishes that AMOTL2 phosphorylation does not affect its mono-ubiquitination. Whether AMOTL2 ubiquitination and phosphorylation are truly linked or independent events, and which signaling molecules mediate this, require further investigation.

## Ubiquitinated AMOTL2 binds to LATS2 and serves as a scaffold for LATS2 activation by SAV1

Thus far, we have established that AMOTL2 mono-ubiquitination by WWP1 promotes LATS activity, which is enabled by the ubiquitin residue that facilitates AMOTL2–LATS2 binding. As proof of concept,

depletion of WWP1 attenuated binding between AMOTL2 and LATS2; this binding was restored by rescue with WT WWP1, and to a much lesser extent by the C890A mutant (Fig 5A). This lends support to the hypothesis that ubiquitinated AMOTL2 directly binds to LATS2.

Although angiomotin proteins have long been established as important upstream components of the Hippo pathway, the biochemical mechanism by which they activate LATS is not known. Interestingly, a recent report suggested that AMOT acts as a scaffolding protein to bridge LATS1/2 together with MST1/2 and SAV1, thereby enhancing LATS phosphorylation and activity (Mana-Capelli & McCollum, 2018). To further demonstrate whether this mechanism holds true in our experimental settings, we assessed the importance of angiomotin ubiquitination in this process. Strikingly, depletion of WWP1 dramatically decreased the interaction of LATS2 with SAV1, an effect that was rescued by ectopic introduction of WT WWP1, but not the WWP1 C890A mutant (Fig 5B). This suggests that WWP1, and its capacity to ubiquitinate AMOTL2, is crucial for the association of LATS2 with its upstream partners. Furthermore, deletion of the UBA domain in LATS2 greatly attenuated binding to SAV1 compared with WT LATS2 (Fig 5C), again highlighting the importance of angiomotin mono-ubiquitination in docking LATS2 and consequently facilitating its activation.

To investigate the effect of Ub-AMOTL2 in greater detail, ubiquitinated AMOTL2 was isolated following in vitro ubiquitination assay of purified AMOTL2 (Fig S9). We then performed LATS2 in vitro kinase assays to assess phosphorylated YAP levels. The addition of purified AMOTL2 increased YAP phosphorylation, an effect that was further enhanced by addition of Ub-AMOTL2 (Fig 5D). To determine whether this increase in YAP phosphorylation was directly dependent on Ub-AMOTL2, we repeated the in vitro kinase assay using increasing amounts of Ub-AMOTL2. When the kinase reaction was performed using WT LATS2, the level of phosphorylated YAP increased in accordance with the amount of Ub-AMOTL2 added. However, when ΔUBA LATS2 (which cannot interact with the ubiquitinated domain of AMOTL2) was used, no such concentration-dependent increase in YAP phosphorylation was seen (Fig 5E). Collectively, these findings indicate that WWP1 overexpression promotes YAP phosphorylation, and that this is a direct consequence of facilitating the interaction between Ub-AMOTL2 and LATS2.

## Contact inhibition leads to WWP1 membrane localization and stabilization

A plethora of upstream cues have been shown to suppress YAP activity, including: serum starvation (Yu et al, 2012), mechanical stress (Dupont et al, 2011), osmotic stress (Lin et al, 2017), and contact inhibition (Zhao et al, 2007; Ota & Sasaki, 2008). Because our previous study assessed the effect of USP9X knockdown and overexpression under sparse and confluent conditions (Kim et al,

for the indicated proteins by Western blotting. **(D, E)** mRNAs from cells prepared as in (D) were analyzed for the indicated genes by qRT-PCR, and expression levels were normalized to those of GAPDH mRNA (n = 4). Data are expressed as means ± SEM (error bars; *P < 0.05, ***P < 0.001; unpaired t test). **(D, F)**. Cells prepared as in (D) were fractionated into cytosolic and nuclear extracts, and analyzed by for the indicated proteins by Western blotting. GAPDH and lamin B serve as controls for the cytosolic and nuclear fractions, respectively. **(D, G)**. Representative images of anti-YAP immunofluorescence in cells prepared as in (D). Green, YAP; blue, DAPI. Scale bar, 50 μm. **(G, H)** Quantification of relative cell populations shown in (G). At least four independent areas were counted. **(I)** 293T cells stably transduced with shGFP (control) or shWWP1 were transfected with the indicated DNAs, then extracts were subjected to LATS2 in vivo kinase assays using bacterially purified maltose-binding protein-YAP as a substrate.

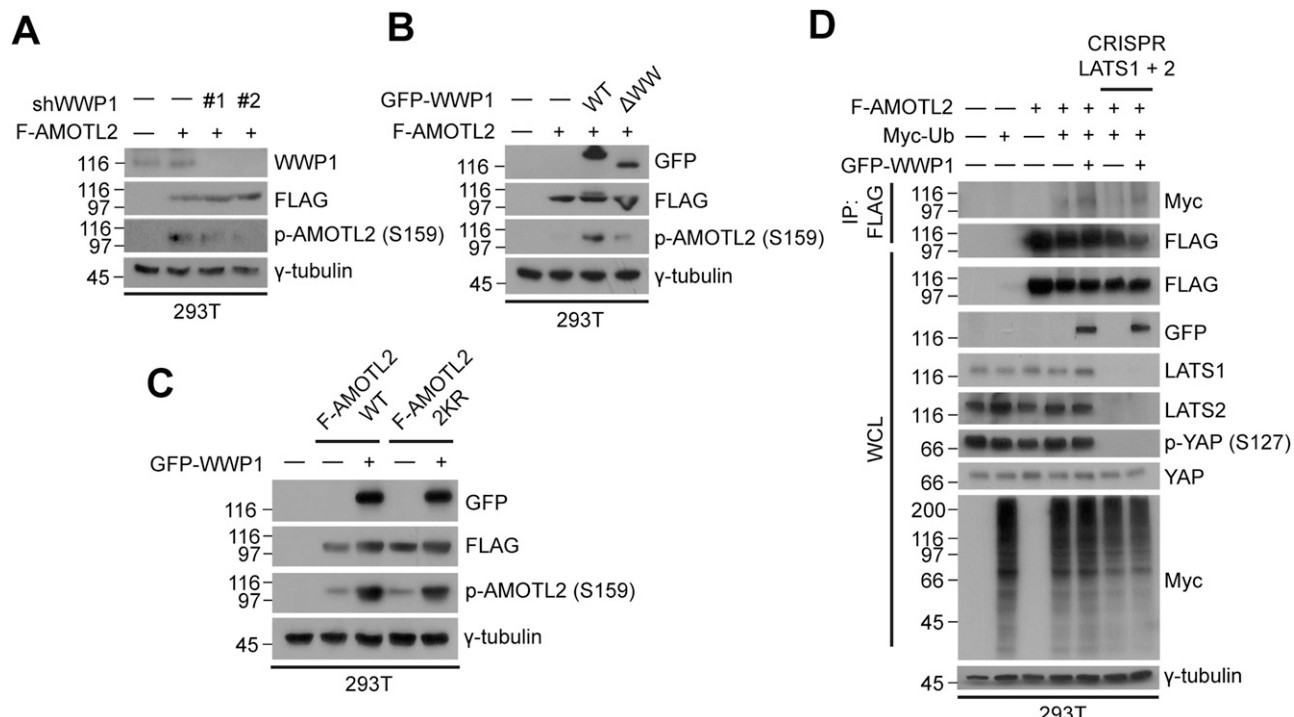

**Figure 4. WWP1 promotes AMOTL2 phosphorylation by LATS.**
**(A)** 293T cells stably transduced with shGFP (control) or shWWP1 lentiviruses were transfected with the indicated DNAs, then lysates were analyzed by Western blotting.
**(B)** 293T cells were transfected with the indicated DNAs, then lysates were analyzed by Western blotting. **(C)** 293T cells were transfected with the indicated DNAs, then lysates were analyzed by Western blotting. **(D)** Control or CRISPR/Cas9-mediated LATS1/2-knockout 293T cells were transfected with the indicated DNAs, then lysates were subjected to in vivo ubiquitination assay.

2016), respectively, we examined whether WWP1 activity is governed by cell density. Indeed, increased mono-ubiquitination of AMOTL2 was evident in confluently cultured cells, but was not detected in cells depleted of WWP1 (Fig 6A).

Although differences in the degree of cell confluence caused no detectable changes in *WWP1* at the mRNA level (Fig 6B), Western blot analyses showed that WWP1 protein levels were dramatically increased under confluent conditions (Fig 6A and C). Depletion of WWP1 confirmed that the increase in the intensity of the WWP1 band detected in Western blots was specific. To test whether this increase in WWP1 protein levels reflected enhanced protein stability, we performed cycloheximide-chase assays under sparse and confluent conditions. This analysis revealed a dramatically greater increase in the half-life of WWP1 in cells cultured at high confluence compared with those cultured under sparse conditions (Fig 6D). Earlier studies on the Hippo pathway, especially those in *Drosophila*, demonstrated that several upstream components, including Kibra, Merlin, Expanded and Warts (KIBRA, NF2, FAT, and LATS1/2, respectively, in mammals), localize to the plasma membrane, and showed that this relocation is critical for their activation (Maitra et al, 2006; Baumgartner et al, 2010). Accordingly, because WWP1 appears to be important for LATS activation under high-density conditions, we tested whether WWP1 exhibits this property as well. Indeed, WWP1 levels at the membrane were increased under high-density conditions, whereas its levels within the cytosol were reduced (Fig 6E). Collectively, these results suggest

that WWP1 is stabilized and mobilized to the plasma membrane upon reaching confluence, a process that serves to reinforce LATS activation.

### WWP1 localizes to the Crumbs polarity complex via AMOTL2

To gain additional insight into how WWP1 is targeted to the plasma membrane, we turned our attention to the Crumbs polarity complex, a critical determinant of apical-basal polarity that functions to regulate tight junctions. Although the complex essentially comprises CRB3, PALS1/MPP5, PATJ/INADL, and MUPP1/MPDZ (Roh et al, 2003; Bulgakova & Knust, 2009), it has been demonstrated that AMOTL2 is a sub-component that is crucial in regulating the apical assembly of the Crumbs complex (Wells et al, 2006; Varelas et al, 2010).

First, we sought to determine whether WWP1 also colocalizes with the Crumbs complex by virtue of its interaction with AMOTL2. As anticipated, whereas membranous localization of wild-type WWP1 was observed and also displayed co-localization with PALS1 at apical junctions, ΔWW WWP1 exhibited a rather diffuse localization throughout the cell (Fig 7A). Moreover, WWP1 was co-immunoprecipitated with the both PALS1 and PATJ, and these interactions were greatly enhanced by supplementation of exogenous AMOTL2 (Fig 7B). Indeed, AMOTL2 was required for WWP1 association with the Crumbs complex, as evidenced by the fact that deletion of the WWP1 WW domain abrogated WWP1 interactions with PALS1 and PATJ (Fig 7C). A similar effect was observed upon expression of the PY[1 + 2m] mutant of AMOTL2 (Fig 7D). These data

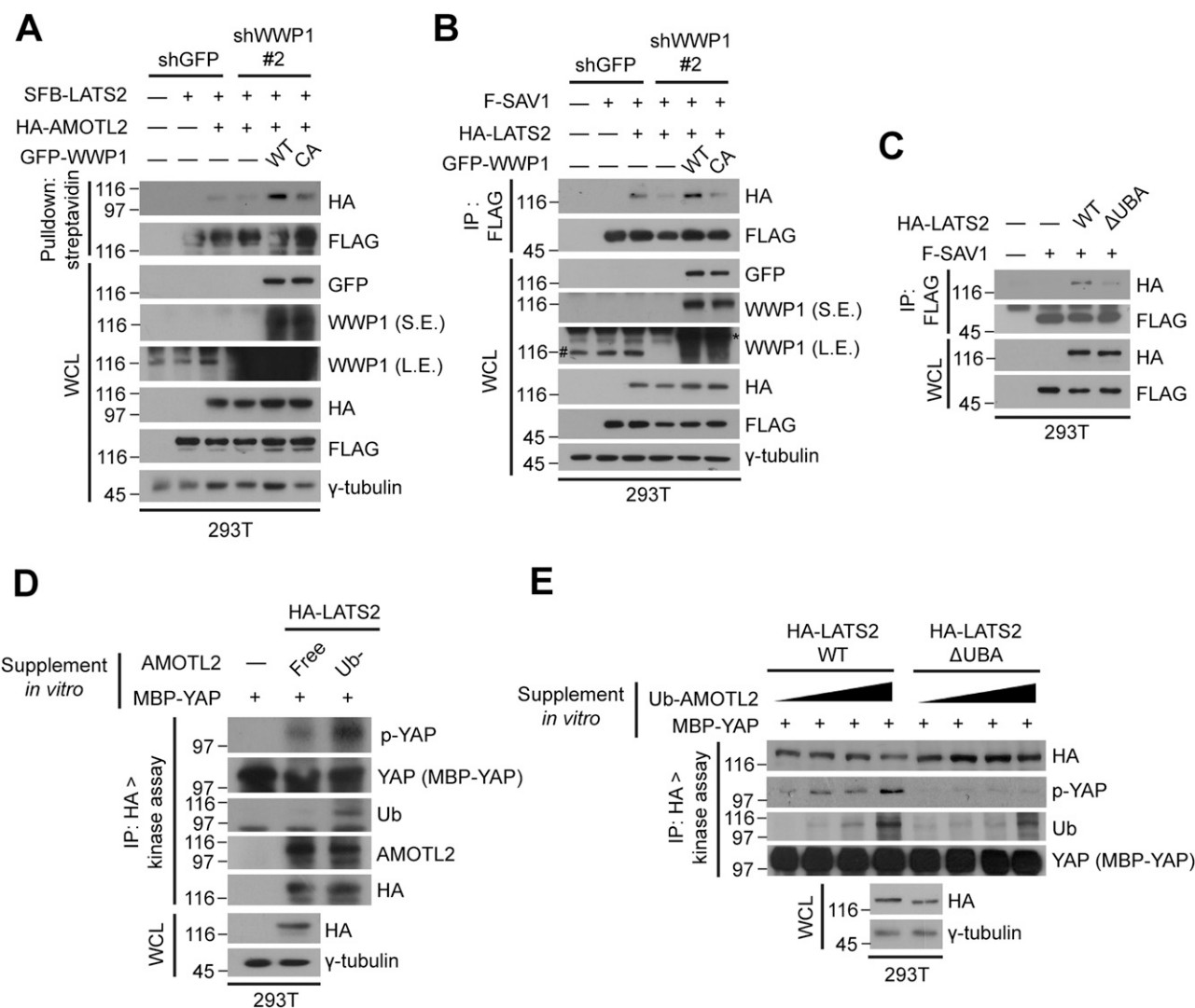

**Figure 5. AMOTL2 mono-ubiquitination promotes LATS activation.**
**(A)** 293T cells stably transduced with shGFP (control) or shWWP1 lentiviruses were transfected with the indicated DNAs, then lysates were pulled down using streptavidin agarose beads. SFB: tag comprising (in order) S-protein tag, FLAG tag, and SBP (streptavidin-binding peptide) tag. **(B)** 293T cells stably transduced with shGFP (control) or shWWP1 lentiviruses were transfected with the indicated DNAs, then lysates were immunoprecipitated with anti-FLAG antibody. #denotes endogenous WWP1 and *denotes exogenous GFP-WWP1. **(C)** 293T cells were transfected with the indicated DNAs, then extracts were immunoprecipitated with anti-FLAG antibody. **(D)** 293T cells were transfected with HA-LATS2, then extracts were subjected to LATS2 in vivo kinase assays using bacterially purified maltose-binding protein-YAP as substrate. Reactions were supplemented with 2 μg of baculovirus-purified free AMOTL2 or Ub-AMOTL2, obtained as a product of in vitro ubiquitination. **(E)** 293T cells were transfected with HA-tagged WT or ΔUBA LATS2 (Ub-acceptor domain deletion mutant), then extracts were subjected to in vivo kinase assays using bacterially purified maltose-binding protein-YAP as a substrate. **(D)** Reactions were supplemented with increasing molar amounts of Ub-AMOTL2 (0, 1, 2.5, and 6 μg), prepared as in (D).

collectively indicate that WWP1 is targeted to the membrane and associates with the Crumbs complex, in particular, via AMOTL2.

If the Crumbs complex is indeed important for WWP1 function at the membrane, loss of Crumbs complex function should prohibit AMOTL2 mono-ubiquitination. Indeed, knockdown of Crumbs3, which tethers the remaining Crumbs complex components to the membrane, dramatically reduced AMOTL2 mono-ubiquitination induced by WWP1 overexpression (Fig 7E). Importantly, depletion of Crumbs3 also greatly attenuated LATS activation under high cell density conditions, as assessed by monitoring mRNA levels of YAP target genes (Fig 7F). Collectively, these data indicate that the Crumbs complex plays a pivotal role in mediating Hippo signaling in response to cell confluence.

In addition, we note that aside from the Crumbs complex, cell-cell contact itself appears crucial for AMOTL2 mono-ubiquitination. Calcium is an important ion which acts as a cofactor for various cell adhesion molecules, and is therefore crucial for the integrity of cellular junctions (Varelas et al, 2010; Mojallal et al, 2014). Upon calcium depletion, we found a dramatic decrease in AMOTL2 mono-ubiquitination (Fig S10A). Interestingly, WWP1 stabilization under high cell density also appeared to be dependent on contact inhibition. MDA-MB-231 triple-negative breast cancer cells are highly invasive and fibroblastic, and continue to clump together even at high confluence (Fig S10B). Surprisingly, stabilization of WWP1 protein did not occur in these cells upon reaching confluence, unlike in MCF10A cells (Fig S10C).

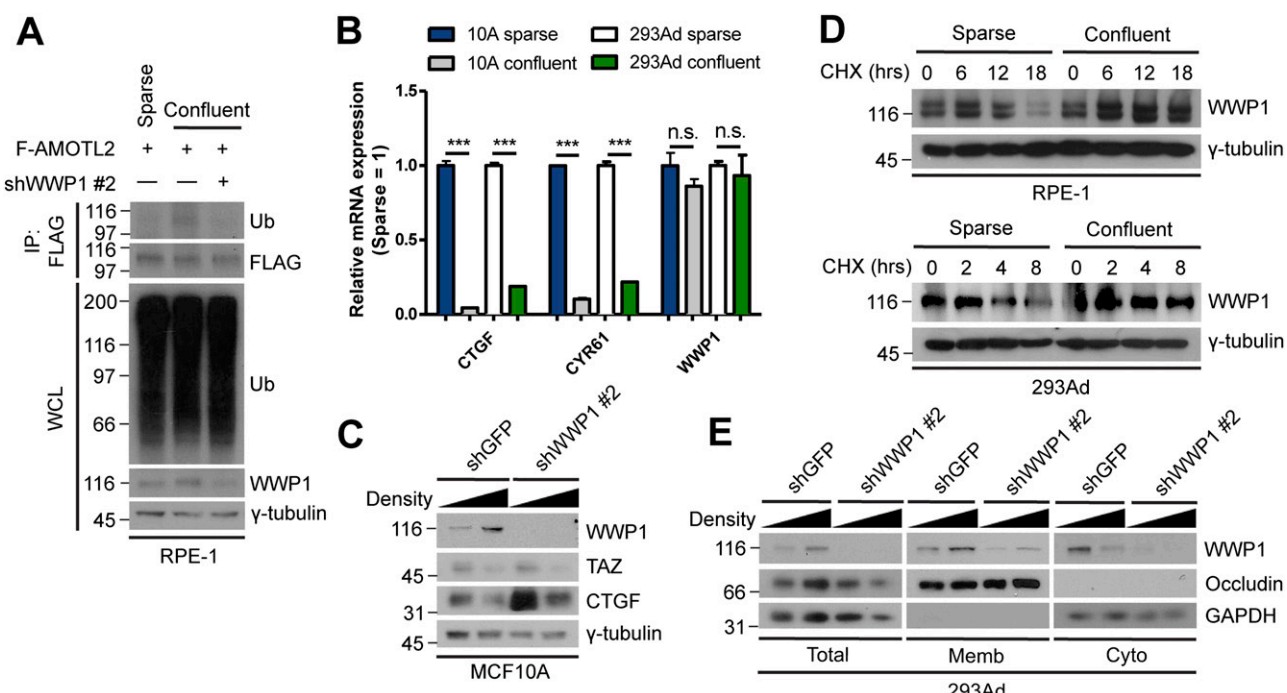

**Figure 6. High cell density promotes WWP1 stabilization and activation.**
**(A)** RPE-1 cells were stably transduced with FLAG-AMOTL2 and either control or WWP1 shRNAs, re-seeded to a sparse or confluent density, then subjected to in vivo ubiquitination assays. **(B)** mRNAs isolated from MCF10A and 293Ad cells, seeded at either a sparse or confluent density, were analyzed for the indicated genes by qRT-PCR, and expression levels were normalized to those of GAPDH mRNA (n = 4). Data are expressed as means ± SEM (error bars; ***$P < 0.001$, n.s. not significant; unpaired $t$ test). **(C)** MCF10A cells stably transduced with shGFP (control) or shWWP1 lentiviruses were seeded at either a sparse or confluent density and extracts were analyzed by Western blotting. **(D)** RPE-1 and 293Ad cells were seeded at either a sparse or confluent density, then a cycloheximide (CHX) chase assay was performed for the indicated times. The resulting extracts were analyzed by Western blotting. **(E)** 293Ad cells were seeded at either sparse or confluent densities and extracts were subjected to S100/P100 membrane-cytosol fractionation assays. The resulting fractions were analyzed for the indicated proteins by Western blotting.

# Discussion

In this study, we characterized the role of the HECT-type E3 ligase, WWP1, in mono-ubiquitinating AMOTL2 based on an unbiased, manual in vivo ubiquitination screening. Importantly, WWP1 ubiquitinates the same lysine residues (K347 and K408) regulated by the deubiquitinase, USP9X, identified in our previous study (Kim et al, 2016). This catalytic reaction requires direct physical binding between WWP1 and AMOTL2 through their respective WW domains and L/PPxY motifs. Functionally, WWP1 overexpression in sparsely cultured epithelial cells (in which YAP is more basally active owing to stronger mechanical tension) resulted in LATS activation and consequently, phosphorylation and nuclear exclusion of YAP and inhibition of YAP-TEAD transcriptional activity. Conversely, WWP1 depletion in confluently cultured cells led to a decrease in LATS activity, as reflected in a reduction in p-YAP, an increase in nuclear YAP, and enhanced YAP-mediated transcription.

It should be noted that a previous study reported an oncogenic role for WWP1 in targeting LATS for proteolytic degradation (Yeung et al, 2013), and high expression levels of WWP1 have been implicated in worse prognostic outcomes in hepatocellular carcinoma (Zhang et al, 2015) and prostate cancer (Chen et al, 2007). Although the present study does not attempt to refute these previous findings, under the conditions tested here, we found no changes in total protein levels of LATS1 or LATS2 (Fig S7), possibly indicating

that the role WWP1 preferentially adopts may be context dependent. Still, this alternate mechanism may explain why the effects of modulating WWP1 levels in cells were not generally striking and only manifest under certain experimental conditions, in this case, differences in cell density. In any case, this once again underscores the importance of WWP1 in regulating LATS, as both positive and negative mechanisms appear to have evolved to fine-tune WWP1 activity.

Controversy also exists as to whether angiomotins activate or inhibit YAP. Although the majority of pioneering studies that characterized the molecular functions of angiomotins asserted its role as a negative regulator of YAP (Chan et al, 2011; Paramasivam et al, 2011; Wang et al, 2011; Zhao et al, 2011), an independent study had reported a cell proliferation-promoting role of AMOT (Ranahan et al, 2011). However, this study focused specifically on the p80 isoform of AMOT, whereas in our study, we used primarily AMOTL2, which is analogous to the p130 (full-length) isoform of AMOT. Importantly, the p80 isoform lacks the L/PPxY motifs at the N-terminus which are required for interaction with WWP1. Given that the PPxY-mutant AMOTL2 failed to be mono-ubiquitinated by WWP1 (Fig S4A), it is likely that any cellular function attributed by AMOT p80 will behave differently from full-length AMOT with respect to regulation of YAP activity. Future in-depth studies characterizing the opposing functions of angiomotin isoforms will be important because the relative expression of protein isoforms can differ significantly depending on the tissue type or cellular context.

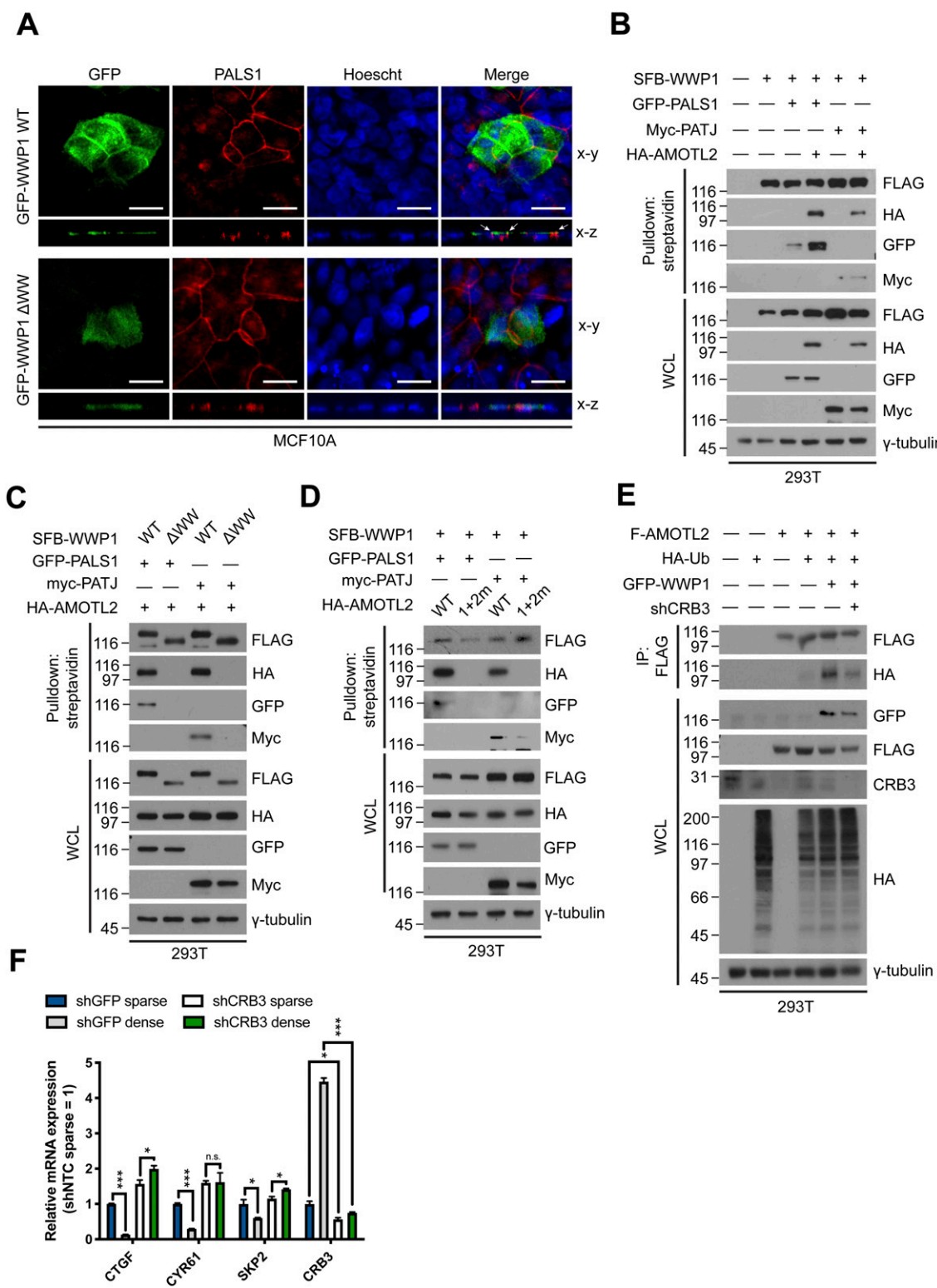

**Figure 7. WWP1 localizes to the Crumbs polarity complex via AMOTL2.**
**(A)** MCF10A cells transfected with GFP-WWP1 WT or ΔWW were immunostained for GFP and PALS1, and serial z-stack images along the x-plane were overlaid by orthogonal projection. Green, GFP; red, PALS1; blue, Hoescht. Scale bar, 50 μm. **(B)** 293T cells were transfected with the indicated combination of DNAs, then subjected to SBP pull-down assays using streptavidin agarose beads followed by Western blot analysis. **(C)** 293T cells were transfected with the indicated combination of DNAs, then subjected to SBP pull-down assays using streptavidin agarose beads followed by Western blot analysis. **(D)** 293T cells were transfected with the indicated combination of DNAs, then subjected to SBP pull-down assays using streptavidin agarose beads followed by Western blot analysis. **(E)** 293T cells were stably transduced with either

Our study provides novel insight into how recruitment of WWP1-AMOTL2 to the Crumbs complex facilitates LATS activation. Upon reaching confluence, WWP1 localizes preferentially to the plasma membrane, where it augments AMOTL2 mono-ubiquitination. Then, by virtue of direct interaction with ubiquitinated AMOTL2, LATS2 is also recruited, becoming activated and subsequently phosphorylating YAP to ultimately attenuate its activity. Through as yet unknown intermediary mechanisms that appear to require the proper establishment of cell polarity in epithelial cells, WWP1 becomes stabilized. This is not a completely unfounded concept, as PTPN14, another upstream regulator of angiomotin-YAP, has also previously been shown to become stabilized under conditions of high cell confluence (Wang et al, 2012b). Although NF2 has often been implicated in responding to contact inhibition through direct interactions with various cadherins and catenins as well as angiomotin (Morrison et al, 2001; Okada et al, 2005), a previous study by Guan and colleagues reported that cell density-dependent LATS activation is likely NF2-independent, although NF2 is absolutely required for responses to serum starvation, a G-protein–coupled receptor–mediated response (Plouffe et al, 2016). Thus, further investigations designed to identify and classify the molecular networks involved in LATS activation upon specific external stimuli will be of critical importance.

Our findings also advance our current understanding of ubiquitination-mediated Hippo signaling, as we directly demonstrated the importance of ubiquitinated AMOTL2 in promoting LATS activity under in vitro conditions. Furthermore, and consistent with our earlier finding that Ub-AMOTL2 functions by acting as a ligand for the LATS UBA domain (Kim et al, 2016), overexpression of WWP1 greatly strengthened the interaction between the two proteins. We also found that WWP1 promotes AMOTL2 phosphorylation at Ser-159, a target site by LATS kinases, although this event is likely independent from AMOTL2 mono-ubiquitination by WWP1. Because this phosphorylation leads to displacement of AMOTL2 from F-actin (Chan et al, 2013; Dai et al, 2013), and therefore increases sequestration of YAP, WWP1 activation leads to a cascade of diverse events that ultimately function to inactivate YAP. However, the biochemical mechanism by which LATS activity is increased upon establishment of a stable interaction with AMOTL2 remains a mystery. According to current understanding, LATS activation requires phosphorylation within its hydrophobic motif by the upstream kinases MST1/2 (Hergovich et al, 2006; Yin et al, 2013) or MAP4K (Meng et al, 2015; Zheng et al, 2015) which causes a conformational shift that promotes auto-phosphorylation (Ni et al, 2015; Mana-Capelli & McCollum, 2018). According to our study, AMOTL2 might serve as an activation platform for MST/SAV1-LATS phosphorylation, given that deletion of the UBA domain of LATS prevented LATS association with SAV1. Importantly, the augmentation of AMOTL2 mono-ubiquitination by overexpressed WWP1 readily increased the association of LATS2 with SAV1. Whether an as yet undiscovered mechanism exists that explains how AMOTL2 promotes LATS activity independent of its regulation by MST is a pressing question that warrants further investigation.

Another critical question that has yet to be fully addressed in the Hippo signaling field is how LATS kinases become activated specifically under high cell density conditions (Fig 8). Upstream junctional polarity complexes, exemplified by Crumbs (Crumbs3-PALS1-PATJ) (Varelas et al, 2010), Par (Par3-Par6-aPKC) (Zhang et al, 2016), and Scribble (Scribble-Lgl-Dlg) (Verghese et al, 2012) complexes, have been shown to recruit LATS to the plasma membrane through mediator molecules. In the present study, we focused on the Crumbs complex because angiomotins are established sub-components of this complex (Wells et al, 2006; Varelas et al, 2010). Crumbs complexes specifically constitute the apical tight junctions (TJs), whose primary function is to serve as a protective or functional barrier to prevent the passage of molecules through the space between adjacent cells (Fanning et al, 1998; Zihni et al, 2016). However, because of their relative specificity, they also serve as signaling modules that facilitate the formation of distinct protein complexes (Itoh et al, 2012; Fredriksson et al, 2015). Of particular interest is the relationship between TJs and Rho signaling because RhoA/ROCK signaling provides a critical input for LATS activation in response to numerous stimuli, including cell density (Ibar et al, 2018), serum starvation (Yu et al, 2012), and mechanical stimuli (Aragona et al, 2013). Because several TJ proteins associate directly with actin (Odenwald et al, 2017) or myosin (Yano et al, 2018), RhoA/ROCK regulation of actomyosin contractility dictates the integrity of cellular junctions (Jou et al, 1998). This junctional status, in turn, can coordinate RhoA/ROCK-mediated LATS activation in a positional manner along the length of the cell by virtue of ubiquitin-mediated AMOTL2–LATS binding. Further characterization of LATS subcellular localization required for its activation would provide tremendous insight into the specific cues affecting the Hippo-YAP signaling pathway.

In summary, our study provides a baseline for understanding the molecular mechanism by which LATS kinases become activated under high cell density conditions. Not only did we show that the E3 ligase WWP1 mono-ubiquitinates AMOTL2 on previously identified sites regulated by the de-ubiquitinase USP9X, we also directly demonstrated the role of this ubiquitin residue in mediating interactions with LATS and causing its activation. We further showed that overexpression of WWP1 in cells suppressed YAP nuclear translocation and activation, and that depletion of WWP1 produced the opposite effects. Importantly, we found that WWP1 itself was stabilized under high-density conditions as a result of membrane localization and interaction with Crumbs complex polarity proteins. Collectively, our findings suggest that the WWP1-AMOTL2 axis is important for cellular responses to increasing confluence, at least in terms of activating the Hippo signaling pathway. Although our study provides only partial evidence concerning the concerted mobilization of the "cell density-Crumbs complex-WWP1-AMOTL2-LATS-YAP" axis in a temporal manner, we nonetheless delineate a genetic relationship and hope that future studies can build upon this framework. Evasion of contact inhibition is one of the defining

control or Crumbs3 shRNAs, then transfected with the indicated DNAs. The resulting extracts were subjected to in vivo ubiquitination assays, followed by Western blot analysis. **(F)** mRNAs isolated from MCF10A cells stably transduced with control or Crumbs3 shRNAs, then re-seeded at a sparse or confluent density were analyzed for the indicated genes by qRT-PCR, and expression levels were normalized to those of GAPDH mRNA (n = 4). Data are expressed as means ± SEM (error bars; *P < 0.05, ***P < 0.001, n.s. not significant; unpaired *t* test).

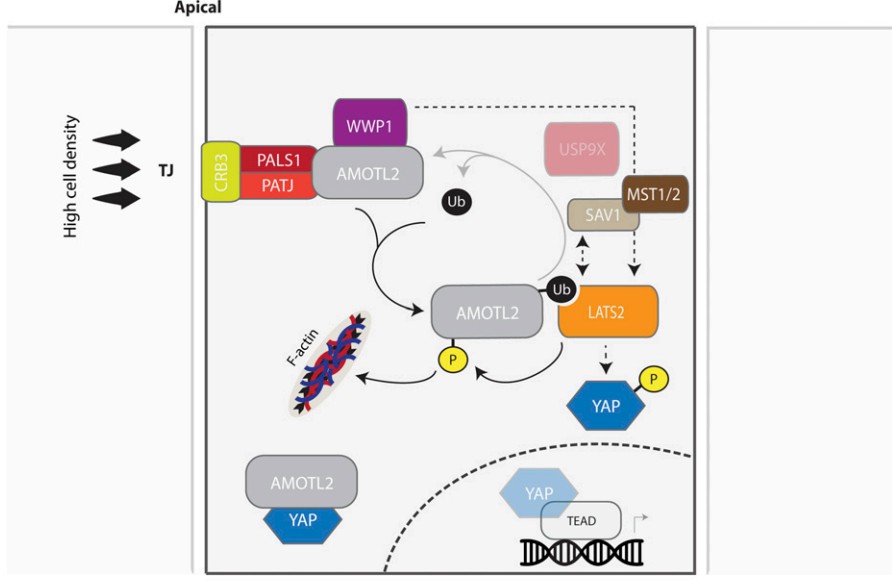

**Figure 8. Model for the regulation and role of WWP1 in the mono-ubiquitination of AMOTL2 and activation of LATS.**
High cell density and the ensuing contact inhibition promotes assembly of the Crumbs complex, which recruits AMOTL2 to tight junctions. By virtue of its association with AMOTL2, WWP1 also preferentially localizes to the plasma membrane, where it becomes stabilized and actively mono-ubiquitinates AMOTL2. Ubiquitinated AMOTL2 then serves as a physical docking site for LATS2, which phosphorylates YAP to promote its cytoplasmic retention and degradation. WWP1 may also activate LATS2 independently from its ubiquitination effect on AMOTL2 (through a yet unidentified pathway), causing AMOTL2 phosphorylation and its displacement from filamentous actin (F-actin), thereby physically sequestering YAP within the cytosol. In summary, contact inhibition activates WWP1 and the ensuing bi-parallel mode of action of WWP1 hinders YAP nuclear entry and activity.

hallmarks of malignant transformation, tumor growth and metastasis; therefore, further understanding of the molecular architecture that governs a cell's response to neighboring cells carries tremendous clinical implications.

# Materials and Methods

## Cell culture

MCF10A cells were cultured in DMEM/F12 (Invitrogen) supplemented with 5% horse serum (Invitrogen), 20 ng/ml EGF (Peprotech), 10 $\mu$g/ml insulin (Sigma-Aldrich), 0.5 $\mu$g/ml hydrocortisone (Sigma-Aldrich), 100 ng/ml cholera toxin (Sigma-Aldrich), and penicillin/streptomycin (Invitrogen). 293T, 293Ad, and MDA-MB-231 cells were cultured in DMEM (Welgene) supplemented with 10% FBS (Invitrogen) and penicillin/streptomycin. RPE cells were cultured in DMEM/F12 supplemented with 10% FBS and penicillin/streptomycin. All cell lines were obtained from American Type Culture Collection, except for 293Ad, which was provided by Dr Kyoung-Jin Oh (KRIBB). Cell lines were validated by DNA fingerprinting of TPOX, TH01, vWA, and D5S818 loci. Cells were routinely tested for the presence of mycoplasma by 4,6-diamidino-2-phenylindole staining.

## Drug treatment and experimental conditions

In cell culture experiments, heclin (Tocris) was used at 20 $\mu$M for 2 h, BrdU (Sigma-Aldrich) was used at 5 $\mu$g/ml for 1 h, and cycloheximide (Sigma-Aldrich) was used at 50 $\mu$M (100 $\mu$M for RPE-1 cells) for the indicated times. For calcium depletion, culture media was replaced with normal (DMEM + no FBS) or $Ca^{2+}$-depleted media ($Ca^{2+}$-depleted SMEM + no FBS) and cultured for an additional 12 h before harvest. Where indicated, sparsely cultured cells were seeded at $3.4 \times 10^3$ cells/cm², and confluently cultured cells were seeded at $10^5$ cells/cm². Experiments were performed 1 d thereafter.

## RNA isolation and analysis

Total RNA was isolated using the easy-Blue reagent (Intron Biotechnology) according to the manufacturer's instructions. Briefly, 2 $\mu$g of RNA was heated at 70°C for 5 min and then reverse-transcribed with M-MLV reverse transcriptase (Enzynomics) using a 1:1 mixture of random hexamer (6-mer) and oligo-dT (18-dT) primers for 2 h at 37°C. The resulting cDNAs were diluted with 1 volume of water and analyzed by quantitative reverse transcription-polymerase chain reaction (qRT-PCR) using a Bio-Rad CFX Connect machine. mRNA levels of target genes were normalized with respect to that of GAPDH. The sequences of primers used are listed in Table S3.

## Immunofluorescence and image acquisition

Cells were seeded on gelatin-coated coverslips at the indicated densities. Cells were then fixed in 4% paraformaldehyde (Sigma-Aldrich) in PBS, permeabilized in 0.1% Triton X-100 in PBS (0.1% PBST), and blocked with 3% BSA in 0.3% PBST. Thereafter, slide-mounted cells were incubated overnight at 4°C with the appropriate primary antibodies, diluted in 0.3% PBST containing 3% BSA. After washing three times with 0.3% PBST, cells were incubated for 1 h at 37°C with appropriate Alexa Fluor-conjugated secondary antibodies (Thermo Fisher Scientific), diluted in PBS. Finally, cells were washed three times with 0.3% PBST and slides were coverslip-mounted using Vectashield mounting medium containing 4',6-diamidino-2-phenylindol (DAPI; Vector Laboratories). Images were collected with an LSM 780 microscope using a C-Apochromat 40×/1.2 W objective lens and ZEN Blue (ver. 3.1) acquisition software (all from Zeiss). For BrdU staining, cells were incorporated with 5 $\mu$g/ml BrdU for 1 h, then fixed with 4% PFA. The fixed cells were further incubated with 2 N HCl at 37°C for 11 min. After three consecutive PBS washes to neutralize the HCl, the cells were processed as above for staining using BrdU antibody. For quantification of YAP subcellular localization or BrdU, at least four independent areas were

counted across three independent experiments, in which at least 100 cells/sample were counted per experiment. For z-stack images, cells were grown on 0.4-µm-pore size polycarbonate Transwell hanging inserts. After fixation, membranes were carefully excised with surgical blades and subsequent immunostaining was performed as described above, except that cells were treated separately with DAPI for 10 min at room temperature after the last secondary antibody wash. After rinsing in 0.3% PBST, membranes were mounted using Vectashield mounting medium without DAPI (Vector Laboratories). For imaging, 50 slices per sample were acquired at 0.5-µm intervals.

### Co-immunoprecipitation

Harvested cells were lysed in NETN lysis buffer (20 mM Tris–HCl [pH 8.0], 100 mM NaCl, 0.5% NP-40, 1 mM EDTA). Cleared cell extracts (1 mg protein in 1 ml) were incubated overnight at 4°C with 1–2 µg of the indicated antibodies. Extracts were incubated with 20 µl of protein A/G agarose beads (GenDepot) for 1 h, after which the beads were washed three times with lysis buffer and boiled with Laemmli buffer. In the case of streptavidin pull-down of SFB-tagged proteins, cleared cell extracts were incubated with 20 µl streptavidin agarose beads (Thermo Fisher Scientific) for 1 h, and then the beads were washed three times with lysis buffer and boiled with Laemmli buffer. The samples were then subjected to SDS–PAGE and Western blot analysis.

### In vivo ubiquitination assay

For assays in 293T cells, cells were transfected with Flag-, HA-, or SFB-AMOTL2 and Myc- or HA-ubiquitin, as indicated, at a 1:3 ratio. For assays in RPE-1 cells, cells were transduced with a Flag-AMOTL2 retrovirus. Harvested cells were lysed with RIPA buffer (50 mM Tris-Cl [pH 7.5], 150 mM NaCl, 1 mM EDTA, 0.5% deoxycholate, 1% NP-40, 0.1% SDS), and then cell extracts (1 mg of protein in 1 ml) were incubated at 4°C overnight with 2 µg of anti-Flag or anti-HA antibody. Extracts were then incubated with 20 µl of protein A/G agarose beads for 1 h, and then the beads were washed five times with RIPA buffer and boiled with Laemmli buffer. In the case of streptavidin pull-down of SFB-AMOTL2, cleared cell extracts were incubated with 20 µl of streptavidin agarose beads (Thermo Fisher Scientific) for 1 h, and then the beads were washed five times with RIPA buffer and boiled with Laemmli buffer. The samples were then subjected to SDS–PAGE and Western blot analysis.

### LATS kinase assay

293T cells transfected with the indicated constructs and/or stably expressing indicated shRNAs were lysed with NETN lysis buffer and the resulting lysate (1 mg of protein in 1 ml) was incubated with 2 µg of anti-HA antibody and processed for immunoprecipitation. In the case of streptavidin pull-down of SFB-LATS2, cleared cell extracts were incubated with 20 µl of streptavidin agarose beads (Thermo Fisher Scientific) for 1 h. Beads were then washed three times with lysis buffer and once with LATS kinase assay buffer (25 mM Hepes [pH 7.4], 50 mM NaCl, 10 mM MgCl$_2$, and 1 mM DTT). Immunoprecipitated LATS2 was incubated for 30 min at 30°C with 1 µg of

recombinant His- or MBP-YAP (purified from *Escherichia coli* BL21 Rosetta strain) and 200 µM of cold ATP, after which immunoprecipitated proteins were denatured by boiling with an equal volume of Laemmli buffer.

### In vitro ubiquitination assay

A mixture containing 50 nM E1, 50 nM UbcH5c, purified GST-WWP1, and 1 U of WT or 2KR mutant AMOTL2 were incubated with 200 mM WT or K0 ubiquitin at 37°C in a reaction buffer containing 50 mM Tris-Cl (pH 7.5), 2.5 mM MgCl$_2$, 2 mM DTT, and 2 mM ATP. Samples were then subjected to SDS–PAGE, and resolved proteins were visualized by Coomassie Brilliant Blue (Bio-Rad) staining and Western blot analysis. Recombinant E1, UbcH5c, and ubiquitin were purchased from Boston Biochem.

### In vitro binding assay

GST or GST-WWP1 (1 µg), immobilized on glutathione-Sepharose 4B beads (GE Healthcare), were incubated with 200 ng of AMOTL2-V5 in binding buffer (20 mM Tris-Cl [pH 7.9], 150 mM KCl, 0.2 mM EDTA, 20% glycerol, 0.05% NP-40, and 0.2 mg/ml BSA) at 4°C for 3 h, after which beads were washed extensively with binding buffer. The beads were boiled with Laemmli buffer, then subjected to SDS–PAGE and Western blot analysis.

### Nuclear-cytoplasmic fractionation assay

Harvested cells were re-suspended in hypertonic buffer (10 mM Hepes [pH 8.0], 1.5 mM MgCl$_2$, 10 mM KCl, and 0.5 mM DTT) and incubated on ice for 20 min. The plasma membrane was disrupted by adding NP-40 to a final concentration of 0.3% and briefly (~10 s) vortexing the samples. Nuclei were pelleted by centrifugation for 1 min, and the supernatant was saved as the cytoplasmic fraction. The nuclear pellet was washed with hypertonic buffer and then extracted with hypotonic buffer (20 mM Hepes [pH 8.0], 25% glycerol, 420 mM NaCl, 1.5 mM MgCl$_2$, 0.2 mM EDTA, and 0.5 mM DTT) for 30 min on ice with frequent vortexing. Nuclear debris was removed by centrifugation for 10 min, and the supernatant was saved as the nuclear fraction. The obtained cytoplasmic and nuclear fractions were boiled with Laemmli buffer, resolved by SDS–PAGE, and then analyzed by Western blotting.

### S100/P100 (plasma membrane-cytoplasmic fractionation) assay

Harvested cells were resuspended in S100/P100 buffer (20 mM Tris-Cl [pH 7.5], 150 mM NaCl, 2.5 mM EDGA, and 1 mM EGTA), and incubated for 20 min on ice. The suspension was then homogenized with a 26G needle and centrifuged at 1,000$g$ to pellet nuclei and unbroken cells. An aliquot of the resulting supernatant, representing the total (membrane + cytosol) fraction, was retained and the remainder of the supernatant was subjected to ultracentrifugation for 1 h at 100,000$g$. The resulting supernatant was collected as the cytoplasmic fraction and concentrated using Amicon microcentricon centrifugal filters (Millipore). The pellet, representing the membrane fraction, was then resuspended in S100/P100 buffer containing 0.1% SDS and 10% glycerol. The obtained fractions were

boiled with Laemmli buffer, resolved by SDS–PAGE, and analyzed by Western blotting.

## Antibodies

The indicated antibodies against the following proteins/peptides were used for Western blot analyses: Flag (F3165; Sigma-Aldrich), HA (MMS-101P; Covance), Myc (sc-40; Santa Cruz Biotechnology), GFP (sc-9996; Santa Cruz Biotechnology), WWP1 (H00011059-M01; Novus Biologicals), WWP2 (A302-935A; Bethyl Laboratories), AMOTL2 (sc-82501; Santa Cruz Biotechnology), USP9X (Abnova, H00008239-M01), CRB3 (NBP1-81185; Novus Biologicals), PALS1 (Santa Cruz, sc-365411), LATS1 (A300-477A; Bethyl Laboratories), LATS2 (#5888; Cell Signaling Technology), p-LATS1 (#8654; Cell Signaling Technology), YAP p-S127 (#4911; Cell Signaling Technology), YAP (H00010413-M01; Novus Biologicals), TAZ (#4883S; Cell Signaling Technology), CTGF (sc-14939; Santa Cruz Biotechnology), CYR61 (sc-13100; Santa Cruz Biotechnology), ubiquitin (550944; BD Biosciences), γ-tubulin (sc-7396; Santa Cruz Biotechnology), occludin (71-1500; Invitrogen), lamin B (sc-6217; Santa Cruz Biotechnology), and GAPDH (ab125247; Abcam); rabbit anti-AMOTL2 p-S159 was a kind gift from Dr Wanjin Hong (National University of Singapore). Antibodies used for immuno-fluorescence analyses were: YAP (H00010413-M01; same as for Western blot; Novus Biologicals), GFP (ab13970; Abcam), and BrdU (555627; BD Biosciences). The antibodies used for immunoprecipitations were the same as those used for Western blot analyses.

## Statistical analysis

Graphs were drawn using GraphPad Prism (ver. 8.1) software. Statistical analyses were performed using two-tailed, unpaired $t$ test with a 95% confidence interval. Statistically significant $P$-values are denoted by asterisks ($*P < 0.05$, $**P < 0.01$, $***P < 0.005$). Unless otherwise stated, all statistically relevant experiments were performed with at least four technical replications and three biological (independent) replications.

# Supplementary Information

# Acknowledgements

We thank Dr Wanjin Hong for kindly providing the rabbit p-AMOTL2 (Ser-159) antibody. This work was funded by a grant from the National Research Foundation of Korea funded by the Ministry of Science and ICT (MSIT) (2020R1A3B2079551 to D-S Lim). D Hwang was supported by the National Research Foundation of Korea (NRF-2014H1A2A1020575).

## Author Contributions

D Hwang: resources, data curation, formal analysis, validation, investigation, visualization, methodology, project administration, and writing—original draft, review, and editing.

M Kim: conceptualization, resources, and data curation.
S Kim: methodology.
MR Kwon: methodology.
Y-S Kang: data curation.
D Kim: validation and writing—review and editing.
H-C Kang: supervision and methodology.
D-S Lim: conceptualization, funding acquisition, project administration, and writing—original draft, review, and editing.

## Conflict of Interest Statement

The authors declare that they have no conflict of interest.

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
