## [Reviewer comments · Life Science Alliance]

Life Science Alliance

AMOTL2 mono-ubiquitination by WWP1 promotes contact inhibition by facilitating LATS activation

Dae-Sik Lim, Daehee Hwang, Miju Kim, Soyeon Kim, Mi Ra Kwon, Ye-Seul Kang, Dahyun Kim, and Ho Kang

DOI: <https://doi.org/10.26508/lsa.202000953>

Corresponding author(s): Dae-Sik Lim, Korea Advanced Institute of Science and Technology (KAIST)

Review Timeline:

Submission Date:	2020-11-04
Editorial Decision:	2021-03-22
Revision Received:	2021-06-21
Editorial Decision:	2021-07-23
Revision Received:	2021-08-03
Editorial Decision:	2021-08-04
Revision Received:	2021-08-06
Accepted:	2021-08-06

Transaction Report:

March 22, 2021

Re: Life Science Alliance manuscript #LSA-2020-00953-T

Prof. Dae-Sik Lim
Korea Advanced Institute of Science and Technology (KAIST)
Biological Science
373-1 Guseong-D. Yuseong-Gu
Yuseong-Gu
Daejeon, Korea 305-701
Korea, Republic of

Dear Dr. Lim,

Thank you for submitting your manuscript entitled "AMOTL2 mono-ubiquitination by WWP1 promotes contact inhibition by facilitating LATS activation" to Life Science Alliance. The manuscript was assessed by expert reviewers, whose comments are appended to this letter. We would like to invite you to submit a revised version of the manuscript that addresses most of the reviewers' points.

We apologize for this extended and unusual delay in getting back to you. As you will note from the reviewers' comments below, while the reviewers are intrigued by these findings, they have raised a number of concerns that need to be addressed prior to further consideration of the manuscript at LSA. In particular, we encourage you to address all of Reviewer 2's points, discuss these findings in the context of Vargas 2020 and Yeung 2013 papers and repeat some of the important experiments in a second cell line to address the concern about 'generality of results' (Reviewer 3 concerns). LOF experiments with CRISPR clones would not be required for publication in LSA, however proper siRNA controls to exclude concerns of off-target effects (raised by Reviewer 3) should be included. All other concerns raised by the reviewers should also be addressed.

Thank you for this interesting contribution to Life Science Alliance. We are looking forward to receiving your revised manuscript.

Sincerely,

Shachi Bhatt, Ph.D.

Executive Editor

Life Science Alliance

<https://www.lsjournal.org/>

Interested in an editorial career? EMBO Solutions is hiring a Scientific Editor to join the international Life Science Alliance team. Find out more here -

https://www.embo.org/documents/jobs/Vacancy_Notice_Scientific_editor_LSA.pdf

B. MANUSCRIPT ORGANIZATION AND FORMATTING:

Reviewer #1 (Comments to the Authors (Required)):

In this manuscript, Hwang et al investigated the crosstalk of E3 ligase WWP1 with AMOTL2, which interacted with LATS2 kinase and ultimately resulted in phospho-YAP cytoplasmic retention and degradation. Specifically, WWP1 mono-ubiquitinates AMOTL2 on two defined lysine residues (K347 and K408). Those are the same lysine residues that the author's group have shown to undergo deubiquitination by USP9X in their previous paper. Ubiquitylated AMOTL2 directly interacts with LATS2 kinase, recruiting SAV1 and promoting YAP phosphorylation. This resulted in YAP cytoplasmic retention. In the context of cellular contact inhibition, high cell confluency induced WWP1 stabilization and membrane localization in cell junctions, interacting with Crumbs polarity complex. This elucidated the activation of Hippo signaling in cellular contact inhibition. I believe that this manuscript is suitable for publication in Life Science Alliance.

One key experiment is to see if WWP1 knockout or knockdown cells are defective in contact inhibition to give more definite conclusion.

Some minor comments:

In the paragraph, "WWP1 promotes LATS activation and thereby inhibits YAP nuclear localization and transcriptional activation", Fig 3D and 3E was mentioned before Fig 3C. The author could rearrange the figures in figure 3 in a chronological order as in the main text.

In Fig 5A, may I know what is "SBP"?

In Fig 5A, gel lane number 4 reflected WWP1 overexpression despite shWWP1 treatment and no GFP-WWP1 transfection. Although the GFP staining showed that WWP1 is most probably not overexpressed, direct staining for WWP1 showed a relatively high expression. What if the author were to show a low exposure (LE) blot, like Fig 5b, to prevent misinterpretation?

Some questions:

The authors established the temporal event that only upon AMOTL2 ubiquitination, AMOTL2 gets phosphorylated by LATS2. It will be interesting to know what is the fate of phosphorylated AMOTL2.

The author deleted the WW domains in WWP1 which ablated interactions with AMOTL2 (Fig 2D). Did the authors delete the domain entirely or replace it with linker sequence? It will be interesting to see what happens if the interaction of WWP1 with AMOTL2 is retained if the WW domains are swapped with that of TAZ or YAP. Nevertheless, the data provided by the author is convincing that WW domain is important for physical interaction with AMOTL2.

Reviewer #2 (Comments to the Authors (Required)):

The manuscript by Hwang et al follows up on prior studies by the Lim group that focused on examining AMOTL2 ubiquitination with this study identifying the ubiquitin ligase WWP1 as a regulator of AMOTL2 mono-ubiquitination. The authors propose that WWP1 levels are increased

with cell density and that WWP1 ubiquitination of AMOTL2 promotes association and activation of LATS2, leading to increased phosphorylation of YAP. They further conclude that the Crumbs complex plays a role in bringing the WWP1-AMOTL2-LATS2 complex together to mediate Hippo signaling.

While WWP1 has been previously identified as a regulator of Hippo signaling, the data in this manuscript suggests that WWP1 function differently than what was previously proposed. The manuscript provides *in vitro* data that generally supports the conclusion that WWP1 promotes AMOTL2 ubiquitination and that this modification promotes the phosphorylation of YAP. While the mechanisms proposed are potentially interesting, there are several deficiencies that reduce enthusiasm. Comments are listed below:

Major comments:

- the authors should comment on why they believe ubiquitinated Amotl2 is observed with the 2KR-AMOTL2 mutant in Fig 1A and 1C, as well as in the control and WWP1-CA1 lanes in Fig 1B.
- it would be helpful to include molecular weight markers in Figure 1B (and any other western blot showing more than a single band) for the reader to better understand what they are looking at.
- The data in the right part of Fig 1C does not appear to align with the protein legend at the top. In particular, the Ub bands do not appear to match the position of AMOTL2, making it difficult to draw conclusions from the figure. Further, the experiments shown in the figure lacks important controls, most importantly showing AMOTL2 without any Ub and E2/E3 enzymes for comparison. In general, what each band represents is unclear in this figure.
- can the authors provide an explanation for why the K0-Ub in Fig 1C shows a smear pattern normally associated with poly-Ub chains in the *in vitro* assay?
- Fig EV3A does not support the conclusion that the WW domain-deleted YAP mutant does not associate with WWP1. The association appears almost the same if the amount of pulled down SFP-YAP is accounted for in the experiment.
- It would be important to show the full length blot with molecular weight markers for Ub in the IP shown in Fig EV3C to strengthen the conclusion that WWP1 mono-ubiquitinates the other AMOT family members. It appears there are additional bands that may have been cropped above what is shown.
- Is the decrease in CTGF, CYR61 and ANKRD1 expression with WWP1 expression shown in Fig 3B statistically significant?
- On Page 6 it is stated that the "WW domain-deleted WWP1 mutant also failed to activate LATS", but no data is linked to this statement.
- The experiment shown in Fig 3I is confusing. It is unclear how did the authors performed an "in vitro kinase assay" together with shRNA-mediated depletion? The methods do not describe these details.
- The data in Fig 3F is not very convincing. The proposed increase in nuclear YAP is minor at best.

The conclusion from this figure could be strengthened by quantitation of replicate experiments.

- The data in Fig EV4C should be quantified. Further, the details for how quantitation was performed in Fig 3H, and how many cells were counted across how many experiments, need to be included.
- On Page 7 it states that WWP1 increases AMOTL2-LATS2 interactions, but this statement is not supported by any data showing this.
- The authors make the conclusion that WWP1 activates LATS. Its phosphorylation of LATS in activating domains increased upon WWP1 expression?
- The changes in p-AMOTL2 with WWP1 expression in Fig 4C is not convincing based on the images shown, particularly when comparing WT and 2KR AMOT. Quantitation of this experiment across replicates may help strengthen the conclusion.
- The data in Fig 5A does not support the statement on page 8 that "depletion of WWP1 attenuated binding between AMOTL2 and LATS2". The binding looks the same between the shGFP and shWWP1.
- Based on what is shown in Fig 5B the lower interaction between LATS and SAV1 with shWWP1 could simply be explained by substantially lower amounts of SAV1 that is present in the WWP1 lane.
- The details for how the Ub-conjugated AMOTL2 was generated is missing. It is unclear what Ub-conjugated means? Is the Ub added to the N-terminus, or was AMOT ubiquitinated in vitro using WWP1.
- It is unclear why the purified AMOTL2 and AMOTL2 conjugated with Ub in Fig 5D run at the same molecular weight in a western blot (mono-Ub should add ~8kDa). The lack of experimental details makes it difficult to assess the data in this experiment.
- It is unclear why WT HA-LATS2 and the delta-UBA HA-LATS2 mutant in Figure 5E are running at the same molecular weight. The UBA domain of LATS2 is relatively large and should significantly change the MW of the protein.
- The data presented for mono-Ub and WWP1 knockdown in Fig 6A is not convincing.
- It is surprising that there is no variation in the expression of CTGF and CYR61 with density across the experiments shown in Fig 6B. The lack of details for what is shown makes this result unconvincing.
- The authors make reference to Fig 4E in the discussion (claiming there are no difference in LATS1 or LATS2 protein), but this figure is absent in the paper. In fact, there is no data examining LATS1, which is the focus of the other paper examining WWP1 and Hippo signaling. This raises the question of whether WWP1 regulates LATS1 and LATS2 similarly, and whether this may account for potential differences observed in this paper versus the prior publication.

Minor comments:

- Details for how many times each experiment was performed should be included in the figure legends.
- The order in how some of the figures are described in the text is not sequential, making it hard for the reader to follow the presentation of the data. For example, Fig EV3 is presented before Fig EV2, Fig 3D is presented before Fig 3C, Fig 3I is presented before 3G, Fig 4D is presented before 4A.
- There are no error bars for WWP1 expression in Fig EV3B and the Y axis scale used cuts off the bars at the top making it difficult for the reader to appreciate the data.
- There are several parts in the text where data is described as "significant" that is not supported by any statistical analysis. The authors should go through the text and make sure that statistics support any significance claims.
- The title of Fig EV3 is "Binding of WWP1 is important for AMOTL2 mono-ubiquitination and subsequent LATS activation", but no data examining LATS is shown in this figure. This title should be revised.
- Additional details should be included for the reagents that are used.

Reviewer #3 (Comments to the Authors (Required)):

In this manuscript, Hwang and colleagues propose that:

AMOTL2 is monoubiquitinated by WWP1

Ub-AMOTL2 interacts with LATS2

Ub-AMOTL2 promotes YAP inactivation

Cell-cell contacts recruit WWP1 together with Crumbs

WWP1/Crumbs complex regulates Ub-AMOTL2

And that this is important when cells form cell-cell contacts to activate LATS1/2 and inhibit YAP/TAZ.

Much work has been already done on AMOTs, cell-cell contacts, and Crumbs in the regulation of YAP, which include different/alternative models to the one proposed here. Moreover, a recent paper (vargas 2020) already made an in depth analysis of AMOT-WWP1 interactions, and provided evidence for the functional role of another WW domain protein, STXBP4, as regulators of YAP in response to cell-cell contacts. Finally, a role for WWP1 as negative regulator of LATS1/2 was previously proposed, but this is opposite to what shown here (yeung 2013). So, the authors should provide more convincing functional evidence that their model has any functional effect, otherwise it is just adding another possible explanation for something we already know (i.e. that cell-cell contacts activate Hippo to inhibit YAP). Please also consider that the functional role of AMOTs remains highly debated, and based in the past either just on gain-of-function or on very correlative evidence, such that any claim on a pro-YAP or anti-YAP function of these molecules should be substantiated by thorough loss-of-function data, which is completely missing here. Finally, it remains completely unknown what is the generality of the findings, since the functional evidence is based on scattered data over two/three cell lines, but without supporting the model in full in at least two independent cell lines (for example, interactions and ubiquitinations are shown in one cell line - independent of cell-cell contacts - while other data in other cell lines).

To this end, it is fundamental that authors work along the following lines:

1) the model that CRB3-WWP1-AMOTL2-LATS1/2 acts as a concerted protein-protein complex in response to cell-cell adhesions must be confirmed experimentally by showing a coherent loss-of-function effect of all the players. This is fundamental to support the proposed model. This must be carefully controlled, because data on RPE1 cells (EV4), for example, are clearly done in non-confluent cells, so that doubts remain over this is a constitutive type of regulation, of something really induced by contacts. As read-outs, since the claim is on contact inhibition of growth, the authors should use both YAP target genes (to show efficient regulation of their transcriptional activity) and proliferation by EdU incorporation, by comparing sparse vs. dense conditions.

2) data on CRB3-WWP1-AMOTL2-LATS1/2 interactions can suffer from the very strong WW-PPXY driven interactions, which enable the formation of these complexes in vitro. Interaction between AMOTs and polarity/junctional complexes is known from a very long time, and these protein-protein interactions have been already reported. However, this is no proof that these complexes exist also in vivo. Co-localization data between WWP1 and Crumbs is very poor - are these cells confluent at all? Are they forming cell-cell adhesions? Is the CRB3/WWP1 signal colocalizing just with the cytoplasm, or with cell-cell adhesion complexes? So, data should be provided to show what factor is required for the localization of the other factors at cell-cell contacts, and to show the relevance of cell-cell contacts for the WWP1-mediated ubiquitination of AMOTL2 (currently lacking).

3) the manuscript suffers from several technical problems that should be solved before any revision.

- The authors claim they isolated WWP1 in a screening, but do not provide data in support. Either show the data in full, or withdraw that claim.
- loss-of-function must be performed with independent shRNA / siRNA sequences, or with independent CRISPR clones. Off targets effects can occur with these techniques, and the authors must exclude them. Moreover, all shRNA sequences including controls (not catalog numbers) must be provided to enable reproducibility of the data (currently missing).
- It remains completely unknown how many independent times the experiments were repeated (were they repeated at all?). Also in the rare cases in which n= is provided, it is not stated whether n is the number of independent biological replicates in a single experiment, or whether the complete experiment was replicated at least two times independently.
- All data on YAP localization must be quantified, and controlled by providing the nuclear density for each sample/condition (to be sure the differences are due to the treatments, as opposed to differences in density).

Our specific responses to the reviewers' comments, indicated in blue, are below.

Reviewer #1 (Comments to the Authors (Required)):

In this manuscript, Hwang et al investigated the crosstalk of E3 ligase WWP1 with AMOTL2, which interacted with LATS2 kinase and ultimately resulted in phospho-YAP cytoplasmic retention and degradation. Specifically, WWP1 mono-ubiquitinate AMOTL2 on two defined lysine residues (K347 and K408). Those are the same lysine residues that the author's group have shown to undergo deubiquitination by USP9X in their previous paper. Ubiquitylated AMOTL2 directly interact with LATS2 kinase, recruiting SAV1 and promoted YAP phosphorylation. This resulted in YAP cytoplasmic retention. In the context of cellular contact inhibition, high cell confluency induced WWP1 stabilization and membrane localization in cell junctions, interacting with Crumbs polarity complex. This elucidated the activation of Hippo signaling in cellular contact inhibition. I believe that this manuscript is suitable for publication in Life Science Alliance. One key experiment is to see if WWP1 knockout or knockdown cells are defective in contact inhibition to give more definite conclusion.

We greatly appreciate the reviewer's favorable response to our manuscript. We agree that the story would be more meaningful and cohesive if the loss of WWP1 leads to a defect in contact inhibition, as is the case for Crumbs3. However, as seen from Figure 6C, WWP1-depleted cells still exhibited LATS activation upon the increase in cellular density, as evidenced by the degradation of TAZ and downregulation of CTGF expression. Therefore, we presume that WWP1 is not solely sufficient to mediate contact inhibitory signals to AMOTL2-LATS, and suspect other players may be involved in this process. For example, other E3 ligases such as RNF14 or ITCH, which based on our ORF library screening data appear capable of mono-ubiquitinating AMOTL2 (which we have now included as Appendix Table S2 and S3, and Fig. EV1), could mediate these signals from the Crumbs complex to AMOTL2-LATS. In fact, it has been previously reported that the loss of NEDD4, another HECT-type E3 ligase which was shown to be stabilized upon high cell density just as in the case of WWP1, abrogated the cell density-dependent activation of the Hippo pathway [Bae et al. *Nat. Comm.* (2015)]. Although the depletion of several E3 ligases simultaneously would be required to provide a definite answer to this, we believe this approach would be impossible to control since AMOTL2 is not the only mono-ubiquitination substrate of these E3 ligases, and not all substrates are shared among them. Therefore, too much variation and off-target effects will hinder from us giving a definitive statement as to which E3 ligases relay contact inhibition to AMOTL2-LATS. Moreover, some E3 ligases such as ITCH have been shown to promote polyubiquitination-mediated proteolysis of LATS, which further hinders accurate analysis. Nevertheless, we agree with the reviewer that this question is mechanistically important, and so future studies dedicated to pursuing the regulation of Hippo-YAP signaling by contact inhibition hold tremendous mechanistic significance.

Some minor comments:

In the paragraph, "WWP1 promotes LATS activation and thereby inhibits YAP nuclear localization and transcriptional activation", Fig 3D and 3E was mentioned

before Fig 3C. The author could rearrange the figures in figure 3 in a chronological order as in the main text.

We greatly appreciate the reviewers' concerns to help improve the cohesiveness and legibility of our manuscript. The main text pertaining to Figure 3 has now been rearranged in a chronological order as suggested.

In Fig 5A, may I know what is "SBP"?

We apologize for our omissions. We now realize that we should have wrote 'SBP' instead of 'streptavidin' in the Figure Legends since we utilize streptavidin-agarose beads to pull down SBP (streptavidin-binding peptide)-conjugated proteins. This text has now been modified, and we thank the reviewer for pointing out this typo.

In fig 5A, gel lane number 4 reflected WWP1 overexpression despite shWWP1 treatment and no GFP-WWP1 transfection. Although the GFP staining showed that WWP1 is most probably not overexpressed, direct staining for WWP1 showed a relatively high expression. What if the author were to show a low exposure (LE) blot, like Fig5b, to prevent misinterpretation?

We thank the reviewer for inspecting our data with great scrutiny and apologize for our shortcomings in clearly portraying our data. Upon close examination, however, we believe that readers should be able to discern the absence of the endogenous WWP1 band in lane #4. Unfortunately, it was difficult to obtain the perfect exposure setting where the endogenous WWP1 band appears clearly absent in lane #4 and yet still present in lanes #1~3 simultaneously, due to the saturation of bands in lanes #5,6 caused by the over-expression of exogenous WWP1. Therefore, we included a lighter exposure strip for the anti-WWP1 blot showing clearly that only the last two lanes depict over-expressed WWP1.

Some questions:

The authors established the temporal event that only upon AMOTL2 ubiquitination, AMOTL2 gets phosphorylated by LATS2. It will be interesting to know what is the fate of phosphorylated AMOTL2.

As discussed within the text, phosphorylation of AMOTL2 at Ser-159 is known to facilitate its displacement from filamentous actin (F-actin), thereby opening the binding interface for YAP. This leads to physical sequestration of YAP by AMOTL2 within the cytosol, ultimately inhibiting its nuclear translocation and transcriptional co-activator function. We are also very curious about the fate of phosphorylated AMOTL2, particularly the dynamics of its phosphorylation/dephosphorylation, as well as the interplay between YAP inhibited due to its direct phosphorylation by LATS and physical sequestration by phosphorylated AMOTL2. It will be important to know whether these independent inhibitory mechanisms cooperate or antagonize one another with respect to YAP-mediated transcriptional activity in the future study.

The author deleted the WW domains in WWP1 which ablated interactions with AMOTL2 (Fig 2D). Did the authors delete the domain entirely or replace it with linker sequence? It will be interesting to see what happens if the interaction of WWP1 with AMOTL2 is retained if the WW domains are swapped with that of TAZ or YAP. Nevertheless, the data provided by the author is convincing that WW domain is important for physical interaction with AMOTL2.

We thank the reviewer for providing this intriguing comment. The Δ WW mutant of WWP1 used in this study is one where the entire domain was deleted. This mutant completely abrogated the interaction with AMOTL2 (Figure 2), however mutation of the most well-characterized L/PPxY motif of AMOTL2 (²¹⁰PPxY⁻²¹³; site necessary and sufficient for interaction with YAP) was insufficient to abrogate interaction with WWP1, which prompted us to search for other L/PPxY motifs. Thus based on these findings, we speculate that replacement of the WW domain of WWP1 with that of YAP or TAZ will likely retain interaction with AMOTL2, albeit with reduced affinity due to the absence of binding interface for other L/PPxY motifs (i.e., ¹⁰⁴LPxY⁻¹⁰⁷) not presumably required for YAP/TAZ-AMOTL2 binding.

Reviewer #2 (Comments to the Authors (Required)):

The manuscript by Hwang et al follows up on prior studies by the Lim group that focused on examining AMOTL2 ubiquitination with this study identifying the ubiquitin ligase WWP1 as a regulator of AMOTL2 mono-ubiquitination. The authors propose that WWP1 levels are increased with cell density and that WWP1 ubiquitination of AMOTL2 promotes association and activation of LATS2, leading to increased phosphorylation of YAP. They further conclude that the Crumbs complex plays a role in bringing the WWP1-AMOTL2-LATS2 complex together to mediate Hippo signaling.

While WWP1 has been previously identified as a regulator of Hippo signaling, the data in this manuscript suggests that WWP1 function differently than what was previously proposed. The manuscript provides in vitro data that generally supports the conclusion that WWP1 promotes AMOTL2 ubiquitination and that this modification promotes the phosphorylation of YAP. While the mechanisms proposed are potentially interesting, there are several deficiencies that reduce enthusiasm. Comments are listed below:

The reviewer provides an accurate synopsis of our findings, and we greatly appreciate the level of scrutiny to our data.

Major comments:

- the authors should comment on why they believe ubiquitinated Amotl2 is observed with the 2KR-AMOTL2 mutant in Fig 1A and 1C, as well as in the control and WWP1-CA1 lanes in Fig 1B.

The reviewer makes a critical point. The most likely explanation as to why AMOTL2 2KR mutant still exhibits residual mono-ubiquitination is that there may exist other residues within AMOTL2 that are capable of being mono-ubiquitinated. That is, even though USP9X de-ubiquitinase may only regulate AMOTL2 mono-ubiquitination at K347/408, WWP1 may mono-ubiquitinate AMOTL2 at yet unidentified residues other than K347/408. If true, then it is likely that these lysine substrates reside outside the coiled-coil domain, since this domain was selected specifically for the identification of mono-ubiquitination sites using mass spectrometry from our earlier study. This discussion has now been added to the main text pertaining to Figure 1 in our revised manuscript. With due respect, the AMOTL2 used in Figure 1B is wild-type AMOTL2, and therefore basal mono-ubiquitination is expected in vector and WWP1 CA-transfected lanes due to the presence of endogenous WWP1.

- it would be helpful to include molecular weight markers in Figure 1B (and any other western blot showing more than a single band) for the reader to better understand what they are looking at.

We thank the reviewer for noting our carelessness. We realize now that due to the nature of our study (which includes many whole blots, multiple proteins, and bands) it would be tremendously helpful to include molecular weight markers in our Western blot images. We have now modified all of our blots to include molecular weight markers.

- The data in the right part of Fig 1C does not appear to align with the protein legend at the top. In particular, the Ub bands do not appear to match the position of AMOTL2, making it difficult to draw conclusions from the figure. Further, the experiments shown in the figure lacks important controls, most importantly showing AMOTL2 without any Ub and E2/E3 enzymes for comparison. In general, what each band represents is unclear in this figure.

As mentioned in reply to the previous comment, we have now included molecular weight markers, which we believe would aid understanding of respective band positions. And as requested by the reviewer, we have cropped a larger area of the *in vitro* ubiquitination assay performed in Figure 1C which shows AMOTL2 without any Ub and E2/E3 enzymes for comparison.

- can the authors provide an explanation for why the K0-Ub in Fig 1C shows a smear pattern normally associated with poly-Ub chains in the vitro assay?

This is a very acute observation, and we understand that smearing of K0-ubiquitin appears at first counter-intuitive. K0-ubiquitin harbors mutations in all seven lysine residues from which multiple ubiquitin chains can arise. However, glycine residues still exist in the C-terminus of ubiquitin that can appear mono-ubiquitinated due to conjugation to substrate lysine residues [please refer to the following literature for additional details: Finley and Chau. *Annu.Rev.Cell.Biol.* (1991), Haas and Siepmann. *FASEB J.* (1997), Wiessman. *Nat.Rev.Mol.Cell.Biol.* (2001)]. However, a single mono-ubiquitination cannot explain the 'smearing' pattern, and therefore we presume multiple mono-ubiquitinations arose in the manner described above.

- Fig EV3A does not support the conclusion that the WW domain-deleted YAP

mutant does not associate with WWP1. The association appears almost the same if the amount of pulled down SFP-YAP is accounted for in the experiment.

We apologize for the poor quality of this figure, making it difficult to draw a definitive conclusion. Thus, we have repeated the experiment to obtain a clearer result. In addition, we also took the reverse approach (i.e. pull down WT or Δ WW WWP1 using wild-type YAP) and found that in this case also, the WW domain-deleted WWP1 mutant was much less efficient in binding to YAP. This result has now become Fig. EV4C (to account for the order in which figures are presented), and the replaced data in which WW domain-deleted YAP fails to associate with WWP1 has become Fig. EV4B.

- It would be important to show the full length blot with molecular weight markers for Ub in the IP shown in Fig EV3C to strengthen the conclusion that WWP1 mono-ubiquitinates the other AMOT family members. It appears there are additional bands that may have been cropped above what is shown.

As mentioned in the previous comment, we have now included molecular weight markers for all Western blot images in our manuscript. With due respect, we believe that the experiment in which WWP1 mono-ubiquitinates other AMOT family members the reviewer is referring to is Fig EV2, not Fig EV3C (which are now Fig. EV3 and Fig. EV4A in our revised manuscript). We have now cropped a larger area of the immunoprecipitated Ub blot for Figure EV3A,B (in addition to Figure EV4A) to show that WWP1 primarily mono-ubiquitinates AMOT family members.

- Is the decrease in CTGF, CYR61 and ANKRD1 expression with WWP1 expression shown in Fig 3B statistically significant?

We have now applied statistics comparing 'Vec vs. WWP1 WT' in Fig 3B. As can be seen, the downregulation of YAP target gene mRNA expression was statistically significant.

- On Page 6 it is stated that the "WW domain-deleted WWP1 mutant also failed to activate LATS", but no data is linked to this statement.

We thank the reviewer for pointing out this potentially misleading statement. The sentence to which the reviewer is referring applies to Fig EV3B, in which we probed for the expression of classical YAP target genes, *CTGF* and *CYR61*. Since YAP activity is primarily governed by its upstream LATS kinases and based on our results, WWP1 regulates YAP via AMOTL2-LATS, we simply inferred that deletion of the WW domains of WWP1 failed to activate LATS (or inactivate YAP). However, since in this case we did not directly assay for LATS activity *per se*, we have now modified the text to claim that "deletion of the WW domains of WWP1 failed to inactivate YAP".

- The experiment shown in Fig 3I is confusing. It is unclear how did the authors performed an "in vitro kinase assay" together with shRNA-mediated depletion? The methods do not describe these details.

We apologize if our explanation of the methods about '*in vitro* kinase assay' was inadequate. In detail, cell lysates from cells transduced with the indicated shRNAs

and transfected with the indicated constructs (excluding MBP-YAP) were immunoprecipitated for the kinase (in this case HA-LATS), and this resulting immunoprecipitate was mixed with bacterially-purified MBP-YAP as a substrate in kinase assay buffer containing ATP to allow for kinase reaction to occur. We have now clarified the Methods section pertaining to this experiment.

- The data in Fig 3F is not very convincing. The proposed increase in nuclear YAP is minor at best. The conclusion from this figure could be strengthened by quantitation of replicate experiments.

We agree with the reviewer that the increase in nuclear YAP intensity shown in Fig 3F appears rather marginal. We have repeated this experiment and replaced the figure with a clearer representative image, which we hope the reviewer finds more convincing. Moreover, to satisfy Reviewer #3's concern of using additional shRNAs to minimize off-target effects, this experiment was also repeated using two independent shRNAs.

- The data in Fig EV4C should be quantified. Further, the details for how quantitation was performed in Fig 3H, and how many cells were counted across how many experiments, need to be included.

As requested, we have quantified the immunofluorescence image for Fig EV4C(which is now Fig. EV5E) . As stated in Figure Legends, at least four independent areas were counted. We have done this across three independent experiments, and counted at least 100 cells per experiment, per sample. This additional detail has now been added to the Methods section pertaining to 'Immunofluorescence and image acquisition'. Moreover, Figures 3G, H have been replaced with experiments conducted using independent WWP1 shRNAs.

- On Page 7 it that states that WWP1 increases AMOTL2-LATS2 interactions, but this statement is not supported by any data showing this.

We thank the reviewer for raising this concern. We realize that our statement that "... WWP1, by virtue of increased AMOTL2-LATS2 interactions, promoted LATS activation..." on page 7 refers to the data presented in Figure 5A, which is presented much later in the manuscript. Therefore, we have now eliminated the interaction-related excerpt within the text to avoid confusion.

- The authors make the conclusion that WWP1 activates LATS. Its phosphorylation of LATS in activating domains increased upon WWP1 expression?

The reviewer poses a critical question. We have primarily relied upon changes in p-YAP (Ser-127) level, a direct phosphorylation site by LATS kinases to gauge 'LATS activity'. To ensure that LATS kinases themselves are indeed activated by WWP1, we measured p-LATS at Thr-1079 (or Thr-1041 for LATS2) (hydrophobic motif; site of phosphorylation by its direct upstream kinases, MST1/2) following pulldown of LATS. As shown below, overexpression of WWP1 showed a marked increase in p-LATS for both LATS1 and LATS2. This new data has now become Fig EV7A,B.

- The changes in p-AMOTL2 with WWP1 expression in Fig 4C is not convincing

based on the images shown, particularly when comparing WT and 2KR AMOT. Quantitation of this experiment across replicates may help strengthen the conclusion.

We understand the reviewer's concern regarding Fig 4C. We therefore repeated this experiment to obtain a clearer result. However, we repeatedly observed that not only WT but also the 2KR AMOTL2 was phosphorylated to a comparable extent upon WWP1 overexpression, in contrast to our earlier claim that the 2KR AMOTL2 exhibited a defect in phosphorylation induced by WWP1. For additional validation, we overexpressed LATS2 Δ UBA along with AMOTL2, and found that this mutant could also promote AMOTL2 phosphorylation as efficiently as WT LATS2 (newly added as Figure EV8A). Collectively, these results indicate that while WWP1 does promote AMOTL2 phosphorylation, this event is likely not mediated by the mono-ubiquitination of AMOTL2 by WWP1 but an independent axis in which WWP1 activates LATS also exists. This aspect definitely warrants further investigation in the future, and the Results and Discussion sections dealing with this data have been modified to reflect this change.

- The data in Fig 5A does not support the statement on page 8 that "depletion of WWP1 attenuated binding between AMOTL2 and LATS2". The binding looks the same between the shGFP and shWWP1.

As the reviewer claims, the purported decrease in the interaction between AMOTL2 and LATS2 does appear rather marginal. However, we respectfully ask the reviewer to consider the fact that re-addition of WWP1 in depleted cells quite significantly increases the association between AMOTL2 and LATS2, which is still in line with our reasoning that mono-ubiquitinated AMOTL2 serves as a docking site for LATS2.

- Based on what is shown in Fig 5B the lower interaction between LATS and SAV1 with shWWP1 could simply be explained by substantially lower amounts of SAV1 that is present in the WWP1 lane.

Similar to the previous concern raised by the reviewer, we respectfully disagree with the claim that the difference in SAV1 amount between shGFP and shWWP1 is so drastic as to be deemed 'substantial'. More importantly, please note that SAV1 expression is rather the same in shWWP1+Vec and shWWP1+WWP1 WT or CA sets, yet there is an apparent restoration of SAV1-LATS2 binding only in the wild-type WWP1 reconstituted lane. Therefore, we believe that our claim that WWP1 ultimately promotes the SAV1-LATS2 association still stands.

- The details for how the Ub-conjugated AMOTL2 was generated is missing. It is unclear what Ub-conjugated means? Is the Ub added to the N-terminus, or was AMOT ubiquitinated *in vitro* using WWP1.

We apologize for our omissions. Although briefly described in the last paragraph pertaining to Figure 5 in the Results section of the text, the 'Ub-conjugated AMOTL2' refers to the AMOTL2 following *in vitro* ubiquitylation reaction with E1, E2, K0-Ub, and WWP1 performed as in Figure 1B. However, in order to isolate AMOTL2 from WWP1, GST-WWP1 was initially bead-bound prior to reaction. Then following *in vitro* ubiquitination, the beads and supernatant samples were separately analyzed by

SDS-PAGE. From this, we could detect ubiquitinated AMOTL2 corresponding to its natural size, devoid of smearing or contamination from WWP1. Therefore, we utilized this supernatant fraction to supplement the LATS kinase assay shown in Figure 5D. So, to answer the reviewer's question, the Ub was not artificially conjugated to AMOTL2 (e.g., to the N-terminus) by any means, but rather is the product of natural mono-ubiquitination by WWP1. We have now newly included this background data as Fig. EV9. In addition, to avoid confusion we have modified the term "Ub-conjugated AMOTL2" to simply "ubiquitinated AMOTL2"

- It is unclear why the purified AMOTL2 and AMOTL2 conjugated with Ub in Fig 5D run at the same molecular weight in a western blot (mono-Ub should add ~8kDa). The lack of experimental details makes it difficult to assess the data in this experiment.

The reviewer makes a very acute point. We believe that the reason why AMOTL2 and Ub-AMOTL2 appear to run at the same size in Figure 5D is since most of the SDS-PAGE gels utilized while preparing this manuscript were rather high percentage (10~12%). This was because in some instances we needed to look at low molecular weight proteins (e.g. CTGF ~37kDa). We hope that our newly added Fig. EV10 sufficiently addresses the experimental details surrounding the generation of Ub-AMOTL2.

- It is unclear why WT HA-LATS2 and the delta-UBA HA-LATS2 mutant in Figure 5E are running at the same molecular weight. The UBA domain of LATS2 is relatively large and should significantly change the MW of the protein.

Similar to the previous comment made by the reviewer, WT and Δ UBA LATS2 appear to run at a similar molecular weight in Figure 5E due to our usage of a higher percentage gel. For clarification, we have re-run these IP samples using a lower percentage gel (7.5%), then immunoblotted for HA. As for the WCL samples, we believe that the shift in molecular weight between WT and Δ UBA LATS2 is already evident.

- The data presented for mono-Ub and WWP1 knockdown in Fig 6A is not convincing.

We agree with the reviewer that the anti-Ub and anti-WWP1 images in Figure 6A appear quite dirty to draw accurate conclusions. We have thus replaced this data with a clearer representative result. Moreover, since the main tenet of Figure 6 is that WWP1 is stabilized upon increasing cellular density, we compared AMOTL2 ubiquitination under sparse and confluent conditions. As can be seen, marked mono-ubiquitination of AMOTL2 was observed only under confluence, which was reversed upon depletion of WWP1.

- It is surprising that there is no variation in the expression of CTGF and CYR61 with density across the experiments shown in Fig 6B. The lack of details for what is shown makes this result unconvincing.

With due respect, we do not fully understand the nature of the reviewer's comment here. Since YAP is known to become phosphorylated and inactivated due to high cell

density, we measured *CTGF* and *CYR61* as readouts for YAP activity in cells plated under sparse or confluent conditions. Perhaps the reviewer is curious as to why the basal (i.e. sparse condition) expression of genes in either cell line is 1.0, and the reason for this is that normalization against *GAPDH* expression was done for each gene, in independent cell lines. As stated in Figure Legends the experiment itself was performed in quadruplicates and the number of cells plated for each condition is provided in the Methods section under 'Drug treatment and experimental conditions', so we do not understand what kind of additional details is required by the reviewer.

- The authors make reference to Fig 4E in the discussion (claiming there are no difference in LATS1 or LATS2 protein), but this figure is absent in the paper. In fact, there is no data examining LATS1, which is the focus of the other paper examining WWP1 and Hippo signaling. This raises the question of whether WWP1 regulates LATS1 and LATS2 similarly, and whether this may account for potential differences observed in this paper versus the prior publication.

We thank the reviewer for pointing out our typo. The reference to Figure 4E in fact refers to Figure 4D, where LATS1 and 2 protein levels are unaltered by WWP1 overexpression (compare lanes 4 and 5). Still, we realize that overall our manuscript currently lacks data to claim that WWP1 promotes the activity of LATS *per se*. As replied to a previous concern raised by the reviewer, we measured p-LATS at the hydrophobic motif (site of phosphorylation by its direct upstream kinases, MST1/2) following pulldown of LATS. As shown below, overexpression of WWP1 showed a marked increase in p-LATS for both LATS1 and LATS2. This new data has now been added to Fig EV7A,B. Moreover, we have newly performed cycloheximide chase experiments to probe for the stability of endogenous LATS1 and LATS2 upon either overexpression of WWP1 in both 293T and MCF10A cells. In either case, LATS protein levels were unaltered under our experimental settings in contrast to the findings obtained by Yeung et al. *PLoS One* (2013), which addressed that WWP1 promotes the proteasomal degradation of LATS1. This new data has also been added as Figure EV7C,D.

Minor comments:

- Details for how many times each experiment was performed should be included in the figure legends.

We had already included the number of times each experiment was performed in cases where statistical analysis was required (e.g. qRT-PCR, BrdU quantification, etc...) as is often the case for papers in which immunoblots are the predominant 'type' of the figure.

- The order in how some of the figures are described in the text is not sequential, making it hard for the reader to follow the presentation of the data. For example, Fig EV3 is presented before Fig EV2, Fig 3D is presented before Fig 3C, Fig 3I is presented before 3G, Fig 4D is presented before 4A.

We thank the reviewer for helping to improve the legibility of our manuscript. This comment has also been addressed by reviewer #1, and we have now rearranged the text so that the figures appear in chronological order.

- There are no error bars for WWP1 expression in Fig EV3B and the Y axis scale used cuts off the bars at the top making it difficult for the reader to appreciate the data.

In fact, upon close inspection there do exist error bars for WWP1 expression in Fig EV3B (which is now Fig. EV4C) albeit quite minor. In regard to the comment on the Y axis, we ask the reviewer to understand our need to separate the Y-axis into two parts as overexpression of WWP1 led to a dramatic increase in *WWP1* mRNA (in this case > 700-fold), and using a singular Y-axis would make it difficult to clearly show the decrease in YAP target genes upon WWP1 overexpression.

- There are several parts in the text where data is described as "significant" that is not supported by any statistical analysis. The authors should go through the text and make sure that statistics support any significance claims.

We apologize for our poor choice of adjectives in describing some of our results, since the word 'significant' should be backed with statistical analysis, especially in the scientific community. We have identified two times in which the word 'significant' was used, once for describing Figure 6C and once in the Discussion "...overexpression of WWP1 significantly strengthened the interaction...". We have now replaced 'significant' with other adjectives in these contexts.

- The title of Fig EV3 is "Binding of WWP1 is important for AMOTL2 mono-ubiquitination and subsequent LATS activation", but no data examining LATS is shown in this figure. This title should be revised.

The fact that activation of LATS *per se* was not thoroughly covered in our original manuscript has been raised several times by the reviewer, and as mentioned above, we have newly added data showing that WWP1 promotes LATS phosphorylation. Nevertheless, this data is not included in Figure EV3 (but later on as Fig. EV7A,B), and therefore we have revised the title of this figure to "Binding of WWP1 is important for AMOTL2 mono-ubiquitination and ultimately YAP inhibition".

- Additional details should be included for the reagents that are used.

With all due respect to the reviewer, we tried to be as thorough as possible in providing sufficient information about the methods and reagents used throughout this study, including the composition of all buffers used, the concentration and duration of which drugs were treated, as well as companies from which the reagents/devices were purchased. If there were specific instances in which the reviewer felt that our explanation was shortcoming, we would greatly welcome the reviewer's input.

Reviewer #3 (Comments to the Authors (Required)):

In this manuscript, Hwang and colleagues propose that:
AMOTL2 is monoubiquitinated by WWP1 Ub-AMOTL2 interacts with LATS2
Ub-AMOTL2 promotes YAP inactivation Cell-cell contacts recruit WWP1 together
with Crumbs WWP1/Crumbs complex regulates Ub-AMOTL2 . And that this is
important when cells form cell-cell contacts to activate LATS1/2 and inhibit
YAP/TAZ.

Much work has been already done on AMOTs, cell-cell contacts, and Crumbs in the regulation of YAP, which include different/alternative models to the one proposed here. Moreover, a recent paper (vargas 2020) already made an in depth analysis of AMOT-WWP1 interactions, and provided evidence for the functional role of another WW domain protein, STXBP4, as regulators of YAP in response to cell-cell contacts. Finally, a role for WWP1 as negative regulator of LATS1/2 was previously proposed, but this is opposite to what shown here (yeung 2013). So, the authors should provide more convincing functional evidence that their model has any functional effect, otherwise it is just adding another possible explanation for something we already know (i.e. that cell-cell contacts activate Hippo to inhibit YAP). Please also consider that the functional role of AMOTs remains highly debated, and based in the past either just on gain-of-function or on very correlative evidence, such that any claim on a pro-YAP or anti-YAP function of these molecules should be substantiated by thorough loss-of-function data, which is completely missing here. Finally, it remains completely unknown what is the generality of the findings, since the functional evidence is based on scattered data over two/three cell lines, but without supporting the model in full in at least two independent cell lines (for example, interactions and ubiquitinations are shown in one cell line - independent of cell-cell contacts - while other data in other cell lines).

We greatly appreciate the constructive criticisms raised by the reviewer, and we tried to overcome many of these shortcomings during the revision process. We agree that the functional role of AMOTs remains highly debated, which reflects the researchers' level of enthusiasm on this topic. As such, by no means are we aiming to resolve this dispute for the last time from our current manuscript. And as the reviewer points out, much work has already been done on AMOTs, cell-cell contacts, and Crumbs on the regulation of YAP. However, we believe that our study is unique in that we identify a novel player within the Hippo-LATS signaling pathway and emphasize the importance of mono-ubiquitination, a post-translational modification not often discussed within this field.

The reviewer also points out a recent paper by Vargas et al., which demonstrated the novel role of STXBP4 as a regulator of YAP. However, this paper does not show how contact inhibition regulates STXBP4, only that it could interact with α -catenin, an actin cytoskeleton-bound protein. In addition, we respectfully disagree with the statement that this paper "made an in depth analysis of AMOT-WWP1 interactions". The paper only notes that WWP1 is a WW domain-containing protein that could interact with AMOT, and do not provide any functional details on the relationship between WWP1 and AMOT since their primary focus was on the role of STXBP4.

As for the comment regarding our usage of different cell lines, we respectfully ask the reviewer to understand that due to the nature of our complex mechanistic study involving multiple components (at times as many as four), we routinely utilized HEK293T cells due to their easily transfect-able nature, high lysate yield (helpful for immunoprecipitation experiments), and versatility. However, these cells are unfortunately not very responsive to various stimuli, including cellular density. Therefore, to assess the role of WWP1 in response to cell density, we performed experiments mainly in MCF10A, supplemented by experiments done in RPE-1 and 293Ad cells. Nevertheless, we believe the main point of our study that AMOTL2 is mono-ubiquitinated by WWP1, is consistent from Figure 6A for which the *in vivo* ubiquitination is performed in RPE-1 cells grown under sparse or confluent conditions. Moreover, in the case of Figure EV4 demonstrating that WWP1 negatively affects YAP activity in RPE-1 cells just as in MCF10A cells (Figure 3), additional data using WWP1 overexpression has newly been added.

To this end, it is fundamental that authors work along the following lines:

1) the model that CRB3-WWP1-AMOTL2-LATS1/2 acts as a concerted protein-protein complex in response to cell-cell adhesions must be confirmed experimentally by showing a coherent loss-of-function effect of all the players. This is fundamental to support the proposed model. This must be carefully controlled, because data on RPE1 cells (EV4), for example, are clearly done in non-confluent cells, so that doubts remain over this is a constitutive type of regulation, of something really induced by contacts. As read-outs, since the claim is on contact inhibition of growth, the authors should use both YAP target genes (to show efficient regulation of their transcriptional activity) and proliferation by EdU incorporation, by comparing sparse vs. dense conditions.

We appreciate the reviewer's suggestion to solidify the generality of our molecular model. With due respect to the reviewer, however, we believe that our manuscript sufficiently demonstrates the linearity of CRB3-WWP1-AMOTL2-LATS-YAP/TAZ signaling module. Firstly, the notion that AMOTL2 is recruited to the Crumbs polarity complex is widely accepted, and is also shown within our manuscript. Then based on our newly observed interaction between AMOTL2 and WWP1, we note that WWP1 is also recruited to the Crumbs complex via AMOTL2. Then, we demonstrate that the depletion of Crumbs3 abrogates the increase in LATS activity due to high cell density (as reflected by the downregulation of YAP target genes), which is in line with the widely accepted fact that loss of LATS abrogates YAP phosphorylation due to high cell density [Plouffe et al. *Mol. Cell* (2016)]. Moreover, we provide sufficient evidence that WWP1 mono-ubiquitinates AMOTL2 and affects YAP activity via LATS. If what the reviewer is demanding is to deplete all of the individual components, then rescue using binding-defective mutants for each component to compare YAP activity under sparse and confluent conditions, we believe this much depth is out of the scope of our study, and believe we have already provided sufficient data to support the genuity of our model.

However, we agree with the reviewer that any functional consequence of our reported WWP1-YAP relationship was missing from the current manuscript. As suggested, we have newly performed BrdU incorporation assay in confluent-cultured cells depleted of WWP1, then rescued using wild-type or catalytically-

defective C890A mutant. In agreement with our earlier results assessing YAP phosphorylation and YAP target gene expression, we found increased cell proliferation upon WWP1 depletion, which was dramatically reversed upon the re-introduction of wild-type, but not C890A WWP1. We have now added this new data into the manuscript as Fig. EV6.

2) data on CRB3-WWP1-AMOTL2-LATS1/2 interactions can suffer from the very strong WW-PPXY driven interactions, which enable the formation of these complexes in vitro. Interaction between AMOTs and polarity/junctional complexes is known from a very long time, and these protein-protein interactions have been already reported. However, this is no proof that these complexes exist also in vivo. Co-localization data between WWP1 and Crumbs is very poor - are these cells confluent at all? Are they forming cell-cell adhesions? Is the CRB3/WWP1 signal colocalizing just with the cytoplasm, or with cell-cell adhesion complexes? So, data should be provided to show what factor is required for the localization of the other factors at cell-cell contacts, and to show the relevance of cell-cell contacts for the WWP1-mediated ubiquitination of AMOTL2 (currently lacking).

We apologize for the poor quality of our data demonstrating the co-localization of WWP1 and Crumbs3. As the reviewer mentions, the interaction between AMOTs and polarity/junctional complexes is already known. However, the point of our study was to show that WWP1, by virtue of its mono-ubiquitination activity on AMOTL2, is also recruited to these complexes, which by itself is a novel statement. One reason why our co-localization data is unclear is due to the poor quality of the CRB3 antibody for immunofluorescence purposes. We therefore purchased and used a PALS1 antibody instead, which showed clear membranous localization at apical surfaces. Moreover, by comparing wild-type and WW domain-deleted WWP1, we could see that the Δ WW mutant exhibited diffuse localization throughout the cell, whereas WT WWP1 also exhibited membrane localization. Figure 7A has now been replaced with this new data.

As to assess whether cell-cell contact is required for WWP1-mediated ubiquitination of AMOTL2, we performed in vivo ubiquitination upon calcium depletion, since calcium is critical for the establishment of cellular junctions. Compared to cells grown under calcium replete conditions, we found that calcium depletion reduced AMOTL2 mono-ubiquitination. Moreover, the ability to sense changes in cell confluence appears to be critical for WWP1 stabilization as MDA-MB-231, which is a highly aggressive and poorly differentiated triple-negative breast cancer cell line that do not exhibit contact inhibition, did not display WWP1 protein stabilization when grown to high cell density. These data have newly been added as Fig. EV10.

3) the manuscript suffers from several technical problems that should be solved before any revision.

- The authors claim they isolated WWP1 in a screening, but do not provide data in support. Either show the data in full, or withdraw that claim.

We apologize for not initially attaching our screening data, and thank the reviewer for pointing this out. As suggested, we have now added our screening results, for which all information can be found in Appendix Table S2 and S3, and Figure EV1.

- loss-of-function must be performed with independent shRNA / siRNA sequences, or with independent CRISPR clones. Off targets effects can occur with these techniques, and the authors must exclude them. Moreover, all shRNA sequences including controls (not catalog numbers) must be provided to enable reproducibility of the data (currently missing).

We appreciate the reviewer's concern to enhance the validity of our loss-of-function data. However, we note that some of our experiments already included independent shRNAs (e.g. Figure 4A, Figure EV1B). Moreover, in some instances when rescue experiments were required (e.g. Figure 3I, Figure 4A,B) we used one shRNA sequence since independent shRNAs target different regions within the ORF, and shRNA-resistant mutants needed to be generated. Nevertheless, we believed it was important to clearly address that loss of WWP1 enhances YAP activity since our manuscript is the first to characterize this relationship in detail. Therefore, we have now replaced all of our loss-of-function data in Figure 3 using independent shRNAs. With due respect to the reviewer, the sequences of all shRNAs and qRT-PCR primers were presented in our original manuscript within the Appendix as Appendix Table S1.

- It remains completely unknown how many independent times the experiments were repeated (were they repeated at all?). Also in the rare cases in which n= is provided, it is not stated whether n is the number of independent biological replicates in a single experiment, or whether the complete experiment was replicated at least two times independently.

We apologize for our lack of statistical details in our original manuscript. We have now amended the Figure Legends and the Methods section to provide more statistical information surrounding our experiments.

- All data on YAP localization must be quantified (Fig EV4C), and controlled by providing the nuclear density for each sample/condition (to be sure the differences are due to the treatments, as opposed to differences in density).

We thank the reviewer for catching this mistake. Fig. EV4C (which has now become Fig. EV5E) has now been quantified.

July 23, 2021

Re: Life Science Alliance manuscript #LSA-2020-00953-TR

Prof. Dae-Sik Lim
Korea Advanced Institute of Science and Technology (KAIST)
Biological Science
373-1 Guseong-D. Yuseong-Gu
Yuseong-Gu
Daejeon, Korea 305-701
Korea, Republic of (South Korea)

Dear Dr. Lim,

Thank you for submitting your revised manuscript entitled "AMOTL2 mono-ubiquitination by WWP1 promotes contact inhibition by facilitating LATS activation" to Life Science Alliance. The manuscript has been seen by the original reviewers whose comments are appended below. You will see that some important issues remain, as outlined by Reviewer 3.

Our general policy is that papers are considered through only one revision cycle; however, we are open to one additional short round of revision. Please note that I will expect to make a final decision without additional reviewer input upon resubmission.

Please respond to Reviewer 3 with another rebuttal letter, and try to find areas to tone down or modify the text, for example in the Discussion, to give more room for some of the criticisms about the detailed nature of the overarching mechanism and how it relates to density.

Please submit the final revision within one month, along with a letter that includes a point by point response to the remaining Reviewer comments.

B. MANUSCRIPT ORGANIZATION AND FORMATTING:

Sincerely,

Reviewer #1 (Comments to the Authors (Required)):

very nicely revised study and i am supportive to publish as it is.

Reviewer #2 (Comments to the Authors (Required)):

The revised manuscript by Hwang et al., which examines the regulation of AMOTL2 by the ubiquitin ligase WWP1, includes several updates that have improved the study. The authors have addressed my original comments and I believe the manuscript is now suitable for publication.

Reviewer #3 (Comments to the Authors (Required)):

1) The main concern I had raised in my revision was that the authors propose a model by which YAP is regulated by cell density, without providing functional evidence in support (i.e. that the players of this model are required to regulate YAP in response to cell density, and that this is relevant to regulate downstream phenotypes such as YAP-driven transcription and YAP-driven proliferation). I also pointed out this was important because some of these players (including WWP1 And AMOTs) have been claimed to operate in very different ways in the Hippo pathway, so that a functional characterization is badly needed to prove their model is correct, as opposed to other models.

The authors dismissed this by stating that "by no means are we aiming to resolve this dispute" over AMOTs function in the Hippo pathway. This is profoundly wrong, as it is on the authors to prove the relevance of their model.

Moreover, they ignored the need to reconcile their data with a previous paper showing the opposite

function of the main "novel" player in the Hippo pathway, WWP1.

Finally, in response to a specific request to show the loss-of-function effects of the main players of the authors' proposed model, they answer "we believe that our manuscript sufficiently demonstrates the linearity of CRB3-WWP1-AMOTL2-LATS-YAP/TAZ signaling module". The issue at stake is not whether the model is linear enough, but whether this has any function in cells.

So, the authors in the end repeatedly refused to provide any additional evidence to solve this issue, and the paper remains without any functional relevance.

2) I had also raised the issue that most of the biochemical data are provided in 293T cells, while other data in other cell lines, so that it remained to be proven that the biochemical data on WWP1 and AMOT ubiquitination explain (at least in part) the phenotypes observed in the other cell lines. The authors state that 293T cells are not responsive to density inhibition of YAP, and that the other cell lines are not amenable to efficient transfection. This problem could have been easily solved by using for example 293 cells which do undergo contact inhibition of YAP (see work by Camargo, Guan, Gumbiner etc.), and which can provide high lysate yield (helpful for immunoprecipitation experiments).

So, the authors did not provide any additional evidence to solve this issue.

4) I had asked to reinforce the notion that AMOT ubiquitination occurs in response to cell-cell contacts. The authors provide biochemical data with calcium depletion in 293T cells. This implies these cells are able to form cell-cell junctions in a calcium dependent manner, which is sufficient to induce their biochemical phenotype. However, the formation of these cell-cell junctions IS NOT sufficient to regulate YAP (as stated by the authors themselves ""HEK293T cells ... are unfortunately not very responsive to various stimuli, including cellular density").

So, this demonstrates that their biochemical phenotype is not relevant to regulate YAP activity.

5) I had raised concerns on the technical quality of the data on cell density, because it was well apparent from the pictures that the authors compared samples at different density (e.g. EV5E). If one compares cells at different cell density, then any result is meaningless. The authors do not provide any new data in support. So, the effects they measure are likely due to errors in plating density, not to a true function of WWP1 and the other players.

Reviewer #1 (Comments to the Authors (Required)):

very nicely revised study and i am supportive to publish as it is.

We thank the reviewer for the favorable response to our revised manuscript.

Reviewer #2 (Comments to the Authors (Required)):

The revised manuscript by Hwang et al., which examines the regulation of AMOTL2 by the ubiquitin ligase WWP1, includes several updates that have improved the study. The authors have addressed my original comments and I believe the manuscript is now suitable for publication.

We are grateful to Reviewer #2 for analyzing our manuscript with great scrutiny, which helped tremendously to improve the overall quality of our manuscript.

Reviewer #3 (Comments to the Authors (Required)):

1) The main concern I had raised in my revision was that the authors propose a model by which YAP is regulated by cell density, without providing functional evidence in support (i.e. that the players of this model are required to regulate YAP in response to cell density, and that this is relevant to regulate downstream phenotypes such as YAP-driven transcription and YAP-driven proliferation). I also pointed out this was important because some of these players (including WWP1 And AMOTs) have been claimed to operate in very different ways in the Hippo pathway, so that a functional characterization is badly needed to prove their model is correct, as opposed to other models.

The authors dismissed this by stating that "by no means are we aiming to resolve this dispute" over AMOTs function in the Hippo pathway. This is profoundly wrong, as it is on the authors to prove the relevance of their model.

Moreover, they ignored the need to reconcile their data with a previous paper showing the opposite function of the main "novel" player in the Hippo pathway, WWP1.

Finally, in response to a specific request to show the loss-of-function effects of the main players of the authors' proposed model, they answer "we believe that our manuscript sufficiently demonstrates the linearity of CRB3-WWP1-AMOTL2-LATS-YAP/TAZ signaling module". The issue at stake is not whether the model is linear enough, but whether this has any function in cells.

So, the authors in the end repeatedly refused to provide any additional evidence to solve this issue, and the paper remains without any functional relevance.

First, we greatly appreciate the reviewer's critical assessment of our manuscript. We mentioned in this manuscript, that the roles of AMOTs and WWP1 in terms of activating the Hippo pathway exhibit some controversy. As for AMOTs, however, the general consensus in the Hippo signaling field is that it acts as a "negative" regulator of YAP, by direct physical sequestration within the cytosol and serving as a molecular platform for binding of upstream negative regulators of YAP (e.g. SAV1) (Mana-Capelli et al. *J. Biol. Chem.*, 2018). The latest Review paper published by the Guan group (Yu et al. *Cell*, 2015), in which the angiomotins were at least mentioned, noted that "... angiomin-like protein 2 (AMOTL2), a negative regulator of YAP...", and any reported role of angiomotins promoting YAP activity were dismissed. Importantly, the original papers published by several groups reported that angiomotins within the Hippo pathway negatively regulate YAP/TAZ (Chan et al., *J. Biol. Chem* 2011, Paramasivam et al. *Mol. Biol. Cell*. 2011, Zhao et al. *Genes Dev*. 2011). They used primarily HEK293/HEK293T and

MCF10A cell lines, which were primarily used throughout our manuscript as well. Thus, it is most likely that under our experimental settings, AMOTs negatively regulate YAP.

A possible explanation for which Ranahan et al. *Cancer Res.* (2011) reported a cell proliferation-promoting role of AMOT is that they focused specifically on the p80 isoform of AMOT. By contrast, we utilized primarily AMOTL2, which is analogous to the p130 (full-length) isoform of AMOT. However, the p80 isoform lacks the L/PPxY motifs at the N-terminus, which are required for interaction of WWP1 (**Reference Figure 1**). Taken together with our result in Figure EV4A which shows that AMOTL2 L/PPxY mutant fails to be mono-ubiquitinated by WWP1, it is likely that any cellular function attributed by AMOT p80 will behave differently from full-length AMOT concerning the regulation of YAP activity. As for the cell proliferation and tumorigenesis-promoting role of AMOT p130 reported by Yu et al. *Sci. Signal* (2013), they initiated their study based on a phenotype arising specifically upon the genetic co-depletion of AMOT and NF2 - another notable negative regulator of YAP within the Hippo pathway – in the liver. This is a profoundly different context compared to our experimental settings, and we think that the biochemical, mechanistic aspect of this paper to explain the pro-YAP role of AMOT is lacking compared to the previous original studies mentioned above.

Reference Figure 1. Functional domains of human angiominin isoforms

Therefore, we are in no position to dismiss these findings and claim that AMOTs only genuinely regulate YAP in a negative manner, but assert that our results and previous literature suggest that AMOTs act in an anti-YAP manner in our study. Importantly, providing a definitive conclusion as to whether AMOTs negatively or positively regulate YAP was not the focal point of our study. The fact that this controversy still exists after 10 years is a testament to the difficulty in addressing the question in a clear, single-handed manner. Although beyond the scope of our manuscript, we believe that future studies dedicated to investigating this discrepancy will provide valuable insights as to any context-specific roles adopted by the angiominins.

With regards to WWP1, we respectfully disagree with the reviewer’s statement that we “... ignored the need to reconcile (our) data with a previous paper showing the opposite function...”. In fact, as requested by the reviewer as well as Reviewer #2, we had newly added data to show that WWP1 promoted LATS1/2 phosphorylation (activity) without affecting their stability under our experimental

conditions (Fig. EV7), in contrast to the pro-degradative role of WWP1 on LATS1 reported by Yeung et al. *PLoS ONE* (2013). For reference, this paper utilized primarily COS-7 cells (not used in our study) for analysis although HEK293T and MCF10A cells were also used for certain experiments. The point is, we failed to notice any LATS proteolytic degradation by WWP1 under our experimental conditions.

Lastly, regarding the comment that we failed to show loss-of-function effects of the main players from our proposed model, we had already addressed this matter in our original responding letter. To review these points again: (1) AMOTs are already widely accepted sub-components of the Crumbs polarity complex (Wells et al. *Cell* 2006, Varelas et al. *Dev. Cell* 2010), and we have also demonstrated the association of AMOTL2 with the Crumbs complex (Fig. 7B-D). (2) Functionally, that is, with respect to ultimately regulating YAP activity, we show that CRB3 depletion led to abrogation of Hippo pathway activation due to high cell density, based on the expression of several YAP target genes (Fig. 7F). (3) The relevance of the “novel” player, WWP1, to this model is that it also associated with the Crumbs complex, by virtue of its pre-existing interaction with AMOTL2 (Fig. 7A-D). The integrity of the Crumbs complex-AMOTL2-WWP1 was altogether important for WWP1-mediated mono-ubiquitination of AMOTL2, since this was dramatically reduced in CRB3-depleted cells. (4) As to whether AMOTL2 regulates YAP activity based on loss-of-function experiments, not only do plenty of literature as outlined above demonstrate this (Chan et al., *J. Biol. Chem* 2011, Paramasivam et al. *Mol. Biol. Cell* 2011, Zhao et al. *Genes Dev.* 2011), but also from our prior study (Kim et al. *EMBO Rep.* 2015), in which identical cell lines and experimental conditions were used for this study. (5) Lastly for LATS1/2, as the direct upstream negative regulators of YAP/TAZ, their relationship is factual to say the least. Also, the Guan group has previously shown that YAP phosphorylation due to high cell density is abrogated in LATS1/2-deleted cells (Plouffe et al. *Mol. Cell* 2016).

As for the “functionality” of our newfound WWP1-AMOTL2 relationship, we have complemented our WWP1 LOF and GOF biochemical data by measuring BrdU+ incorporation in WWP1 depleted cells, which were then rescued with shRNA-resistant wild-type or catalytically-inactive mutants. In line with our findings that WWP1 inhibits YAP activity in a catalytic activity-dependent manner (Fig. 3), we found that WWP1 suppressed cell proliferation (specifically, entry of the cell-cycle into the S phase) in a catalytic activity-dependent manner (Fig. EV6). Therefore, we respectfully disagree with the reviewer’s statement that “... the paper remains without any functional relevance”.

2) I had also raised the issue that most of the biochemical data are provided in 293T cells, while other data in other cell lines, so that it remained to be proven that the biochemical data on WWP1 and AMOT ubiquitination explain (at least in part) the phenotypes observed in the other cell lines. The authors state that 293T cells are not responsive to density inhibition of YAP, and that the other cell lines are not amenable to efficient transfection. This problem could have been easily solved by using for example 293 cells which do undergo contact inhibition of YAP (see work by Camargo, Guan, Gumbiner etc.), and which can provide high lysate yield (helpful for immunoprecipitation experiments).

So, the authors did not provide any additional evidence to solve this issue.

With respect to the reviewer, we believe that this point had also been addressed previously in the original responding letter. The reviewer noted that “it remains to be proven that the biochemical data on WWP1 and AMOT ubiquitination... in the other cell lines”, and we believe that this point has been demonstrated in Fig. 6A (for which we assessed AMOTL2 mono-ubiquitination under sparse or confluent density as well as WWP1 dependency in RPE-1 cells) and Fig. EV10A (for which AMOTL2 mono-ubiquitination was assessed in 293Ad cells in a manner akin to biochemical experiments performed in 293T cells). Furthermore, we had already demonstrated in our prior study (Kim et al. *EMBO Rep.* 2015) that cells grown to high density exhibit increased AMOTL2 mono-ubiquitination (see Fig. 5C) in RPE-1 cells. The Guan lab extensively utilizes HEK293A cells for their biochemical studies (Plouffe et al. *Mol. Cell* 2016, Park et al. *Cell* 2015), a derivative of HEK293 cells specialized for the production, amplification, and titrating of replication-incompetent adenovirus. 293Ad cells, which were used in our manuscript are also derivatives of parental HEK293 cells specialized for

adenovirus production. It appears that the nomenclature of these cells differs according to the commercial vendor (HEK293A from Thermo Fischer; 293Ad from Cell Biolabs) from which they originated. Although the reason for discrepancies in cellular response to various stimuli is unclear, we have found that 293Ad cells retain response to cell density, serum, and actocytoskeletal disruption (as opposed to parental HEK293 and 293T cells; **Reference Figure 2**), and therefore routinely use these cell lines for cases in which cellular competence for response to stimuli are required.

Reference Figure 2. Response of HEK293 (parental), 293T, and 293Ad cells to LPA stimulation following overnight serum starvation, as measured by immunoblotting for p-YAP (S127)

4) I had asked to reinforce the notion that AMOT ubiquitination occurs in response to cell-cell contacts. The authors provide biochemical data with calcium depletion in 293T cells. This implies these cells are able to form cell-cell junctions in a calcium dependent manner, which is sufficient to induce their biochemical phenotype. However, the formation of these cell-cell junctions IS NOT sufficient to regulate YAP (as stated by the authors themselves "HEK293T cells ... are unfortunately not very responsive to various stimuli, including cellular density").

So, this demonstrates that their biochemical phenotype is not relevant to regulate YAP activity.

We apologize for this mislabeling and thank the reviewer for pointing it out. As can be seen from the bottommost label Fig. EV10A, this experiment was performed in 293Ad cells, which retain responsiveness to a variety of signals including cell-cell contact (Fig. 6B,E). These cells are also calcium-responsive (Kon et al. *Cell Death Discov.* 2017, Serrat et al. *PLoS ONE* 2013), and still provide a relatively high lysate yield, which was why they were chosen for the immunoprecipitation experiment following calcium depletion. However, we noticed that in the Figure Legends pertaining to this experiment we had mistakenly written 293T instead of 293Ad cells, a typo likely caused by copy pasting the Legend description of similar experiments (e.g. Fig. EV4A, Fig. EV9D). We have once again meticulously looked over all of our Figure Legends to ensure that details are in line with the Figures themselves. Once again, we apologize for our carelessness, and thank the reviewer for noticing this.

5) I had raised concerns on the technical quality of the data on cell density, because it was well apparent from the pictures that the authors compared samples at different density (e.g. EV5E). If one compares cells at different cell density, then any result is meaningless. The authors do not provide any new data in support. So, the effects they measure are likely due to errors in plating density, not to a true function of WWP1 and the other players.

Although we appreciate the reviewer's concern as to the consistencies of our experiments, we ask the reviewer to re-assess our results with greater scrutiny. Taking a look at Fig. EV5E, as pinpointed out

by the reviewer, for example, we count roughly 23 DAPI+ cells in shGFP, and 20 DAPI+ cells in shWWP1#1 from this representative image. Thus, we believe that a difference of 2~3 cells per field (~10%) is small enough to be accredited for stochastic experimental variation, and we assure that the same number of cells per sample were plated for each experiment. Importantly, unlike MCF10A cells that grow to a honeycomb-like monolayer upon reaching confluence (Fig. EV10B), RPE-1 cells do not necessarily, due to variation in cellular morphology and size. Therefore, it is difficult to manually ensure that cells are distributed evenly as in MCF10A cells. Nevertheless, we decided to append a second representative micrograph for each sample from this experiment, as well as show uncropped images encompassing a greater number of cells per field to lend more credibility (**Reference Figure 3**).

** Dotted red box is for editor/reviewer's eyes only

Reference Figure 3. The revised figure for Fig. EV5E now includes two representative images per sample, including uncropped images from the original figure (outlined in dotted red box)

August 4, 2021

RE: Life Science Alliance Manuscript #LSA-2020-00953-TRR

Prof. Dae-Sik Lim
Korea Advanced Institute of Science and Technology (KAIST)
Biological Science
373-1 Guseong-D. Yuseong-Gu
Yuseong-Gu
Daejeon, Korea 305-701
Korea, Republic of (South Korea)

Dear Dr. Lim,

Thank you for submitting your revised manuscript entitled "AMOTL2 mono-ubiquitination by WWP1 promotes contact inhibition by facilitating LATS activation". We would be happy to publish your paper in Life Science Alliance pending final revisions necessary to meet our formatting guidelines.

- please consult our manuscript preparation guidelines <https://www.life-science-alliance.org/manuscript-prep> and make sure your manuscript sections are in the correct order
- please add your table legend to the main manuscript text after the main and supplementary figure legends
- please upload Table S1 separately with file designation "Table" instead of as Appendix
- LSA allows supplementary figures, but no EV Figures; please update your callouts for the Supplementary Figures in the manuscript Fig EV1A=Fig S1A; while supplementary figures use the system supplementary Fig S1;
- please note that titles in the manuscript file and the system must match
- please add callouts for Figures S5A-F and S6A-C to your main manuscript text
- Is there a splice between the last two columns in Figure 4C, second row (FLAG)? If so, please indicate it.

LSA now encourages authors to provide a 30-60 second video where the study is briefly explained. We will use these videos on social media to promote the published paper and the presenting author. Corresponding or first-authors are welcome to submit the video. Please submit only one video per manuscript. The video can be emailed to contact@life-science-alliance.org

A. FINAL FILES:

B. MANUSCRIPT ORGANIZATION AND FORMATTING:

Sincerely,

August 6, 2021

RE: Life Science Alliance Manuscript #LSA-2020-00953-TRRR

Prof. Dae-Sik Lim
Korea Advanced Institute of Science and Technology (KAIST)
Biological Science
373-1 Guseong-D. Yuseong-Gu
Yuseong-Gu
Daejeon, Korea 305-701
Korea, Republic of (South Korea)

Dear Dr. Lim,

Thank you for submitting your Research Article entitled "AMOTL2 mono-ubiquitination by WWP1 promotes contact inhibition by facilitating LATS activation". It is a pleasure to let you know that your manuscript is now accepted for publication in Life Science Alliance. Congratulations on this interesting work.

DISTRIBUTION OF MATERIALS:

Again, congratulations on a very nice paper. I hope you found the review process to be constructive and are pleased with how the manuscript was handled editorially. We look forward to future exciting

submissions from your lab.

Sincerely,
